# Impacts on air dose rates after the Fukushima accident over the North Pacific from 19 March 2011 to 2 September 2015

**Kuo-Ying Wang**[1]*, **Philippe Nedelec**[2], **Hannah Clark**[3], **Neil Harris**[4], **Mizuo Kajino**[5], **Yasuhito Igarashi**[6]

**1** Department of Atmospheric Sciences, National Central University, Chung-Li, Taiwan, **2** Laboratoire d'Aérologie, Centre National de la Recherche Scientifique, Observatoire Midi-Pyrénées, Toulouse, France, **3** IAGOS-AISBL, 98 Rue du Trône, Brussels, Belgium, **4** Centre for Environment and Agricultural Informatics, Cranfield University, Cranfield, United Kingdom, **5** Meteorological Research Institute (MRI), Japan Meteorological Agency (JMA), Tsukuba, Ibaraki, Japan, **6** Division of Nuclear Engineering Science, Institute for Integrated Radiation and Nuclear Science, Kyoto University (KURNS), Osaka, Japan

* kuoying@mail.atm.ncu.edu.tw

**Data Availability Statement:** The data underlying the results presented in the study are available from http://doi.org/10.5281/zenodo.3563214.

## Abstract

A fleet of thirteen in-service global container ships continuously measured the air dose rates over the North Pacific after the Fukushima Daiichi Nuclear Power Station (FDNPS) accident. The results showed that the elevated air dose rates over the Port of Tokyo and the FDNPS emissions are significantly correlated (log(emission fluxes) = 54.98 x (air dose rates) (R = 0.95, P-value<0.01), and they are also significantly correlated with the Tsukuba deposition fluxes (log(deposition fluxes) = 0.47 + 30.98 (air dose rates) (R = 0.91, P-value<0.01). These results demonstrate the direct impact of the FDNPS emissions on the depositions of radionuclides and the air dose rates over the Port of Tokyo. Over the North Pacific, the correlation equations are log(emission fluxes) = -2.72 + 202.36 x (air dose rates over the northwestern Pacific) (R = 0.40, P-value<0.01), and log(emission fluxes) = -0.55 + 80.19 x (air dose rates over the northeastern Pacific) (R = 0.29, P-value = 0.0424). These results indicate that the resuspension of the deposited radionuclides have become a dominant source in the transport of radionuclides across the North Pacific. Model simulations show underestimated air dose rates during the periods of 22-25 March 2011 and 27-30 March 2011 indicating the lack of mechanisms, such as the resuspension of radionuclides, in the model.

## Introduction

The safety of nuclear power plants is a major issue and a significant health concern for the general public [1, 2]. There were 33 serious accidents associated with nuclear power plants worldwide during a 60-year period spanning 1952–2011 [3]. This statistic indicates that there was approximately one serious accident every 22 months. While the existence of a

**Funding:** KYW is funded by the Ministry of Science and Technology (www.most.gov.tw; 107-2111-M-008-027-). PN and HC are supported by the the IAGOS project (www.iagos.org). The funders had no role in study design, data collection and analysis, decisionto publish, or preparation of the manuscript.

**Competing interests:** The authors have declared that no competing interests exist.

nuclear power plant is a voluntary risk to the host country, the transport of contaminated particles in the air comprises an involuntary risk for neighboring countries and the global environment. At this occurrence rate, it is necessary to systematically and continuously collect data on the atmospheric concentrations of radionuclides and the air dose rates for the risk assessment and management associated with nuclear power plants. The main characteristic of a nuclear accident is that the source of the nuclear accident is highly localized, but the effect ripples around the globe. For example, the Chernobyl nuclear accident occurred on 26 April 1986. Ground-level monitoring data revealed elevated levels of atmospheric concentrations of radionuclides over northern Europe after 26 April 1986. Model simulations showed that high levels of radionuclides from the Chernobyl nuclear power plant accident were transported across Europe by atmospheric winds [4, 5]. On 11 March 2011, a magnitude 9.0 earthquake occurred at a distance of approximately 154 km northeast of the Tokyo Electric Power Company (TEPCO) Fukushima Daiichi Nuclear Power Station (FDNPS; $37.420°N$ and $141.033°E$; see Fig 1 for the epicenter of the 11-March-2011 earthquake, and the FDNPS site) [6]. The earthquake produced a 13-meter-high tsunami in the FDNPS area, inundated the power plant with seawater, knocked out electricity to the primary water cooling pumps, and destroyed 12 of the 13 emergency diesel generators, resulting in the overheating and explosion of the boiling-water reactors and the release of radioactive nuclear materials into the atmosphere [7]. The collapse of the FDNPS was another wake-up call illustrating the need for the assessment of risks associated with the operation of nuclear power plants [8].

After the FDNPS accident, measurements of radioactive nuclides and air dose rates were made in the air, in the drainage close to the FDNPS, and in the surface seawater of the North Pacific [9–12] to monitor the impacts of the spatial-temporal dispersion of radioactive materials. However, no continuous measurements of the air dose rates over the North Pacific have been reported. In this work, we present continuous measurements of the air dose rates over the North Pacific and the northwestern Atlantic atmospheres after the FDNPS accident from a fleet of thirteen in-service global container ships and from Port of Tokyo sites with the Pacific Greenhouse Gases Measurement (PGGM) project [13]. Continuous measurements of the air dose rates were collected from 294 cruises with 41,477 sampling locations and 248,864 measurements during a period from 19 March 2011 to 2 September 2015. The PGGM data were compared with the estimated emissions (rate) from the FDNPS, the measured deposition (rate) in Tsukuba city in Japan, the measurements in the drainage close to the FDNPS, and a model simulation.

In the following sections, we document the sensors and the methods used in this work (Section 2). Section 3 describes the monitoring results over the Port of Tokyo, in the North Pacific atmosphere and in the northwestern Atlantic atmosphere from March 2011 to September 2015. These monitoring data represent the spatial and temporal distribution of the air dose rates from a local area (Port of Tokyo) up to a hemispheric scale (North Pacific and the northwestern Atlantic) after the FDNPS accident. The impact of the FDNPS accident on the atmosphere is studied with a linear regression model, which is used to associate the FDNPS emission fluxes (in Bq/h of $^{137}Cs$, with a half-life of approximately 30 years, and $^{134}Cs$, with a half-life of approximately 2 years) and the measurements of the deposition fluxes at Tsukuba [14] with the air dose rates. The FDNPS emission fluxes were estimated monthly from March to September 2011 [15] and from October 2011 to August 2015 [16]. Finally, we used the air dose data reported in this work to verify a model simulation of the atmospheric dispersions of radionuclides released by the FDNPS [17].

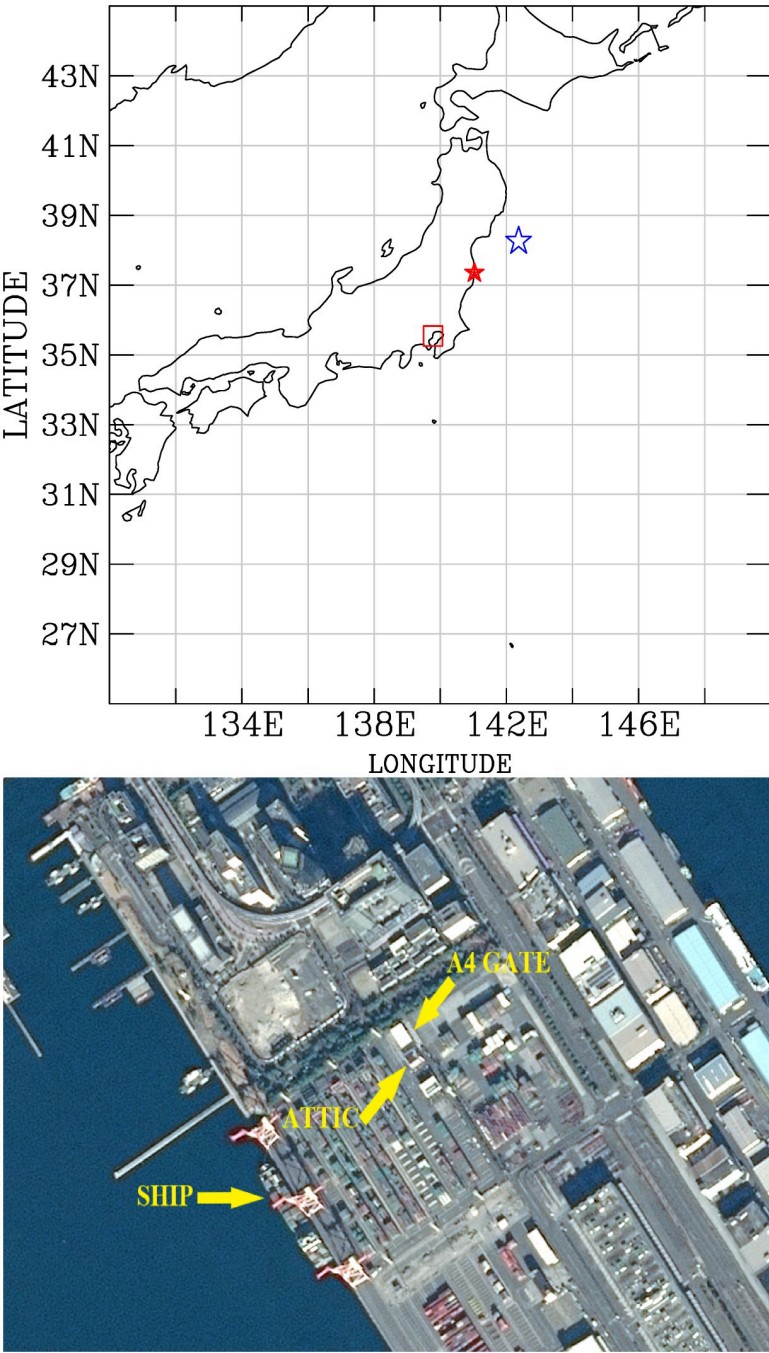

**Fig 1. Sites of air dose measurements.** (Upper panel) Locations for the measurements of the radioactivity dose rates at the Port of Tokyo site (open red square), 11-March-2011 earthquake epicenter (38.322°N; 142.369°E; open blue star), and the FDNPS (solid red star). Map made with Natural Earth. Free vector and raster map data at naturalearthdata.com. (Lower panel) A magnified satellite image (Map data: With permission from CSRSR under the CC BY 4.0 license), showing the attic area in the Port of Tokyo office (ATTIC; 35.6150°N; 139.7806°E); A4 gate (GATE; 35.6150°N, 139.7806°E), and location for calling ships (SHIP; 35.6121°N, 139.7783°E).

## Data and methods

### The PGGM monitoring platform over the North Pacific

As soon as we saw the news of the FDNPS accident on TV, we anticipated subsequent atmospheric transport and dispersion of contaminated nuclei from the FDNPS over the North Pacific regions. What happened over Europe after 26 April 1986 because of the Chernobyl nuclear power plant accident [5] could take place over the Pacific areas. The PGGM project equipped a fleet of nine global container ships from the Evergreen Marine Corporation (EMC) with carbon dioxide ($CO_2$) analyzers that have been acquiring $CO_2$ data for climate research since 2009 [13, 18]. This good long-term collaboration with the EMC enabled us to quickly conceive a project to immediately monitor the air dose rates over the North Pacific regions after 19 March 2011. The quick response and flexibility associated with the EMC made the urgent deployment of monitoring devices for radioactivity dose rates possible. Table 1 shows a list of 14 sensors that were deployed at the Port of Tokyo and 13 container ships on 2011–2-15 to collect the air dose rates.

### The monitoring devices and calibrations

Radioactivity dosimeters from Thermo Scientific RadEye B20 and RadEye G10 were used in this work. The upper limit of the gamma dose rate measurement of the B20 sensor is 2 mSv/h [19–22]. The upper limit of the measurement of the G10 sensor is 100 mSv/h [23–25]. Both the B20 and G10 sensors use an energy compensated GM-tube detector. Table 1 shows a list of the instrument types, serial numbers, locations, and observational periods. Table 1 shows the lowest calibration values (10 $\mu$Sv/h), calibrated results, and calibration dates for all fourteen sensors used in this work. The first calibrations occurred during March-April 2011, before the sensors were deployed to the land-based site and container ships. The second calibrations took place on 6 November 2015 when all the sensors were removed from the container ships and land-based site and returned to National Central University for preparations. The sensors have an uncertainty [26, 27] of less than ±3%.

In the calibration of the sensors used in this work, shown in S1 Appendix of S1 Fig, we collected all the measured air dose rates versus the designated air dose rates. From this data set,

**Table 1. List of instrument number (No), type, serial number (SN), measurement sites, observational periods, calibration values for 10 $\mu$Sv/h dose rate, and the calibration dates.** Cal-1 means first calibrations, and Cal-2 means second calibrations.

| No | Type | SN | Site | Observational Period | Cal-1 | Cal-2 | Cal-1 Date | Cal-2 Date |
|----|------|-----|------|---------------------|-------|-------|-----------|-----------|
| G01 | B-20 | 01059 | Tokyo | 19/03/2011 − 02/09/2015 | 9.55 | 10.14 | 01/04/2011 | 06/11/2015 |
| G02 | B-20 | 01009 | Container Ship | 11/04/2011 − 28/11/2014 | 9.55 | 10.12 | 17/03/2011 | 06/11/2015 |
| G03 | G-10 | 02113 | Container Ship | 23/03/2011 − 03/05/2015 | —- | 10.36 | — | 06/11/2015 |
| G04 | G-10 | 02111 | Container Ship | 08/04/2011 − 26/03/2015 | 10.56 | 10.64 | 24/03/2011 | 06/11/2015 |
| G05 | G-10 | 02120 | Container Ship | 30/03/2011 − 06/06/2015 | 9.99 | 10.30 | 24/03/2011 | 06/11/2015 |
| G06 | G-10 | 02118 | Container Ship | 30/03/2011 − 05/04/2015 | 9.97 | 9.96 | 24/03/2011 | 06/11/2015 |
| G07 | G-10 | 02115 | Container Ship | 04/04/2011 − 05/03/2015 | 10.14 | 10.10 | 24/03/2011 | 06/11/2015 |
| G08 | G-10 | 02114 | Container Ship | 04/04/2011 − 22/12/2014 | 10.01 | 10.60 | 24/03/2011 | 06/11/2015 |
| G09 | G-10 | 02119 | Container Ship | 06/04/2011 − 25/01/2015 | 10.12 | 9.87 | 24/03/2011 | 06/11/2015 |
| G010 | B-20 | 00993 | Container Ship | 12/04/2011 − 25/01/2015 | 10.00 | 10.24 | 17/03/2011 | 06/11/2015 |
| G011 | B-20 | 01060 | Container Ship | 15/04/2011 − 29/03/2015 | 8.86 | 9.36 | 08/04/2011 | 06/11/2015 |
| G012 | B-20 | 01054 | Container Ship | 20/04/2011 − 24/04/2015 | 9.56 | 9.80 | 08/04/2011 | 06/11/2015 |
| G013 | B-20 | 01055 | Container Ship | 26/04/2011 − 09/08/2014 | 9.92 | 10.34 | 08/04/2011 | 06/11/2015 |
| G014 | B-20 | 01053 | Container Ship | 23/04/2011 − 14/03/2015 | 9.05 | 9.94 | 08/04/2011 | 06/11/2015 |

we calculated the differences between the measured air dose rates versus the designated air dose rates. We then ranked these differences. The 50[th] percentile of the differences (between the measured versus the designated air dose rates) are within 2% of the designated air dose rates. The 75[th] percentile of the differences (between the measured versus the designated air dose rates) are within 5% of the designated air dose rates. For each sensor, the calibration versus designated air dose rates are shown in S1 Appendix of S1 Fig The differences between the calibrated and designated values are sensor dependent. For example, sensor numbers G01 and G11 show large discrepancies in the high dose rates compared with the low dose rates; sensor number G04 shows low discrepancies in the high dose rates compared with the low dose rates.

The sensors have an operating temperature range of $-20°C$ to $50°C$ [28]. The calibrations of all the sensors were measured against a PTW32002 1-liter spherical ionization chamber (PTW TM32002, SN: 298; PTW-Freiburg, Freiburg, Germany) radiation detector [29] at the Nuclear Science and Technology Development Center at National Tsing Hua University.

$^{137}Cs$ was the radioactive source for the sensor calibrations, at strengths of 11 GBq, 18.5 GBq, and 1850 MBq (the $^{137}Cs$ sources were obtained on 1 July 1996). Calibrations were conducted at four designated dose rates for each sensor: 10 $\mu$Sv/h, 80 $\mu$SV/h, 200 $\mu$SV/h, and 800 $\mu$SV/h. The calibration results for the 10 $\mu$Sv/h dose rate are shown in Table 1. Detailed calibration results against the four designated dose rates for each sensor are shown in S1 Appendix of S1 Fig in the Supporting Information. The calibration results were analyzed with a linear regression model [30, 31] that compares the designated (given) dose rates with the measured dose rates for each sensor. The analysis shows that the calibration factors, determined as the ratios of the measured dose rates to the calibrated dose rates, of each B20 and G10 sensor are consistently within the 1% (50[th] percentile) and 4% (75[th] percentile) deviations from the designated dose rates that are lower than 100 $\mu$Sv/h. Large deviations exist for air doses higher than 100 $\mu$Sv, with the largest discrepancies occurring at 800 $\mu$Sv/h. Some sensors perform remarkably well (within 5% of the designated dose rates) from 10 $\mu$ Sv/h to 800 $\mu$Sv/h (packages G03, G05, G06, G07, G09, and G10).

The calibration factors for the four designated air dose rates (10, 80, 200, and 800 $\mu$Sv/h) are calculated as the ratios of the designated air dose rates to the measured air dose rates (squares and crosses in the right-hand side panels for each sensor shown in S1 Appendix of S1 Fig). From these four calibration factors, we use a linear regression model [30, 31] to find the calibration factors $y$ as a function of the air dose rates of $y = A + bx$. Here, $y$ is the calibration factor, $x$ is the air dose rate, A is the intersection in the y-axis, ($x = 0$), and b is a fitting coefficient describing the changes in $y$ with respect to changes in $x$. The straight lines on the right-hand side of each panel of S1 Appendix of S1 Fig) show the results of $y$ for each sensor at the two calibration times. The measured air dose rates on the container ships and at the Port of Tokyo sites are less than 0.50 $\mu$Sv/h, which is very close to the y-axis when x = 0. Hence, we calculate the calibration factors as $y = A$. Factor A for the first calibration in 2011 is A1. The second calibration in 2015 is A2. Both A1 and A2 are calculated for each sensor. We then build a look-up table of calibration factors, Ai, for each day of measurements by linearly interpolating A1 and A2 in time. This look-up table Ai is then used in the calibrations of the measured dose rates for each B20 and G10 sensor. The results and distribution of Ai for each sensor are shown in the bottom panel of S1 Appendix of S1 Fig We note that the calibration factors A1 and A2 are close to the calibration factors for 10 $\mu$Sv/h, as shown in S1 Appendix of S1 Fig

## The monitoring sites

Sensor number G01 was used to measure the outdoor air dose rates at the Port of Tokyo sites (Fig 1). The measurements were taken at three locations one-by-one at each sampling time in

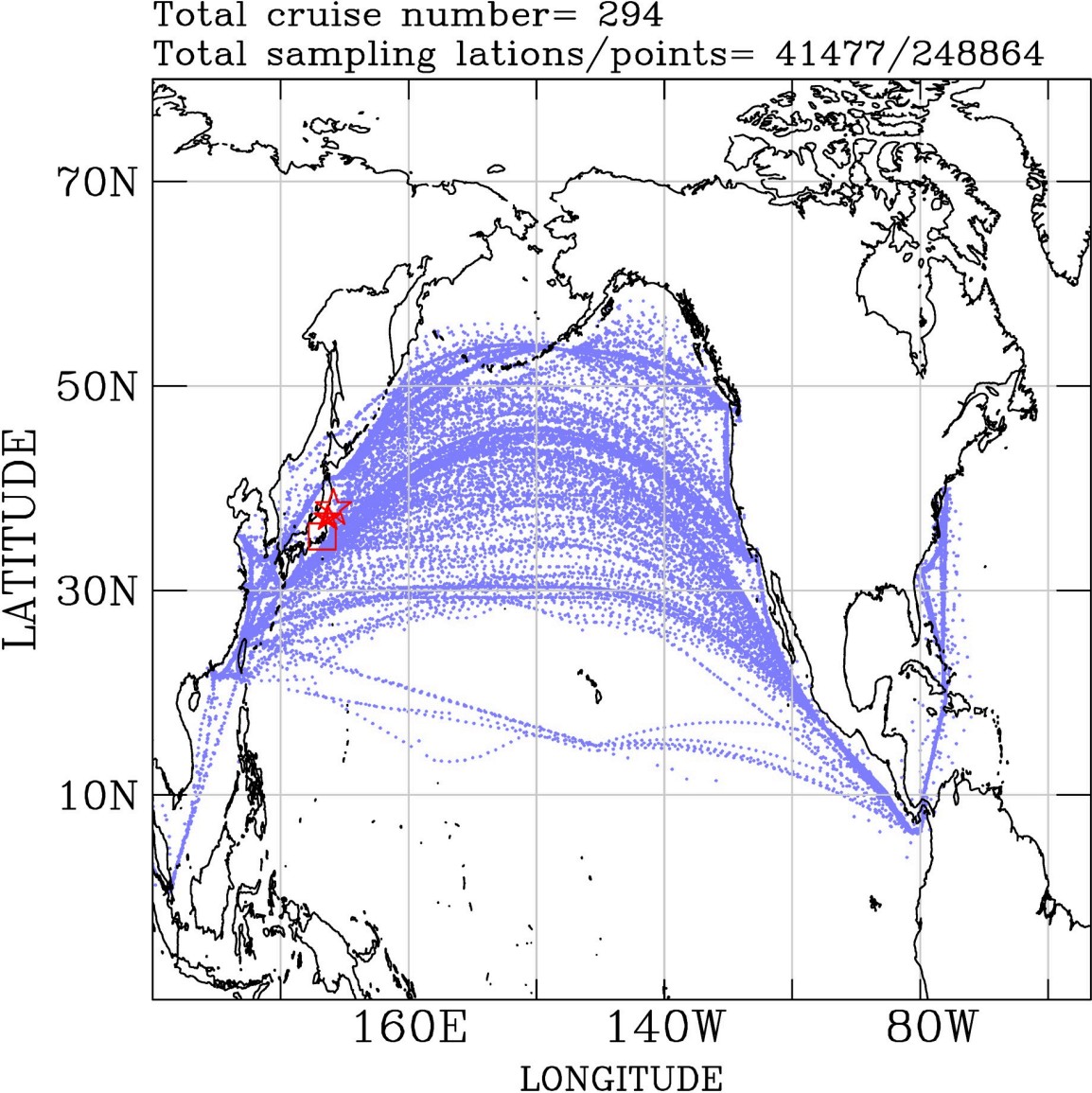

**Fig 2. Measurements of the radioactivity dose rates over the North Pacific and the northwestern Atlantic marine atmosphere.** The open star is the location of the 11-March-2011 earthquake epicenter, the solid star is the FDNPS site, and the open square is the Port of Tokyo office area. Map made with Natural Earth. Free vector and raster map data at naturalearthdata.com.

the attic of the office building (35.6150˚N, 139.7806˚E), the A4 terminal gate (35.6150˚N, 139.7806˚E), and the calling ships (35.6121˚N, 139.7783˚E; this same site for multiple calling ships). All three locations were sampled sequentially in time with the same G01 sensor and within the same sampling hour twice daily during the working day. The counting time for one measurement of the air dose rate was 10 minutes at each site. We note that the measurements were logged as hourly results. The Port of Tokyo is located 228 km southwest of the FDNPS. The remaining thirteen sensors were deployed aboard container ships operating over the North Pacific and the northwestern Atlantic (Fig 2). Table 1 shows a list of air dose sensors and measurement periods.

## Data collection and validation

Measurements were taken every 6 hours onboard the ships. These measurements were made manually by officers of the ship at the compass deck, which is located on the upper deck of the ship. (approximately 40 meters above sea level, depending on the type of ship and the loading of the containers). We note that X-band (9.419 GHz) and S-band (3.050 GHz) radars are used on the container ships. The radar was installed at a vertical distance between 7 and 11 meters above the compass deck (depending on the ship type). Both radars have no effects of electromagnetic waves on the gamma dose rate sensors used in this work. The readings were taken from 6 directions: east, west, north, south, upward, and downward. Measurements at the Port of Tokyo were also manually taken at 00 UT and 06 UT hours during the working day. The readings were taken in the same six directions as those onboard the ships. The six directions were used to account for the contributions of the air dose rates from all possible directions. The counting time for each direction was approximately 1 minute when the sensor readings were stabilized. To test the sensitivity of the measurements as a function of distance from the ground, two additional measurements were taken: at 50 cm and 10 cm above the ground in the attic of the office. There are three calibrated files in the G01 data folder: A4, outdoor, and vessel. 'A4' represents the measurements made at the gate. 'Outdoor' represents the measurements made in the office attic. 'Vessel' represents the measurements on the calling ships over the water at the Port of Tokyo. The outdoor file contains measurements related to heights at 50 cm and 10 cm from the ground. The height-related measurements are shown in the last three columns of data (DOWN, 50 cm, and 10 cm) in the outdoor files.

## Emission and deposition fluxes

The previous sections described more than 4 years of time-series measurements of the air dose rates over the Port of Tokyo, the North Pacific Ocean, and the northwestern Atlantic Ocean. Additional releases of the radionuclides to the atmosphere from the FDNPS continuously occurred after the accident [15, 16]. The long-term measurements of the monthly deposition fluxes of radionuclides were collected at Tsukuba, Japan [14]. In this work, we use a linear regression model [31] to determine if there are associations between the air dose rates (measured over the Port of Tokyo, the North Pacific Ocean, and the northwestern Atlantic), the Tsukuba deposition fluxes and the FDNPS emission fluxes. The statistical significance of the associations was tested with chi square test [30].

## Model simulations

Models are essential tools for evaluating the global dispersion of radionuclides from the FDNPS in the atmosphere [32, 33]. Model simulations were verified against the amounts of airborne radionuclides deposited on the ground (in units of $Bq/m^2$), and suspended in the air (in units of $Bq/m^3$) [16, 32, 34]. Both deposited and airborne radionuclides contribute to air dose rates [34, 35]. As shown in the following sections, the air dose rates measured over the land surface of the Port of Tokyo were due to the contribution of the ground deposited and air suspended radionuclides. The air dose rates measured over the oceans on the Port of Tokyo calling ships, the North Pacific Ocean, and the northwestern Atlantic Ocean sailing ships were due to airborne radionuclides. Hence, measurements of the air dose rates provide other methods for verifying the dispersion simulations of the FDNPS radionuclides.

In this work, we used the atmospheric transport and dispersion model (ATDM) results organized by the World Meteorological Organization (WMO) to assess atmospheric dispersion following the FDNPS accident [32, 36–39]. Simulations of the dispersion of the radioactive clouds were conducted by using the Lagrangian transport method [38]. A unit source

represented by 100,000 particles/h was released from the FDNPS during each 3-hour interval [17, 39] from 11 March 2011 at 00 UT to 3 April 2011 at 2100 UT [40]. The model has a domain area from $125°E$ to $155°E$ in longitudes, and from $28°N$ to $48°N$ in latitudes. The model contains 601x401 longitude-latitude grids with a regular 0.05° grid resolution (approximately 5 km), a vertical resolution of 100 m, and centered at $38°N$ and $140°E$. The in-cloud scanvenging of the particles is parameterized by a wet scavenging rate of $3 \times 10^{-5} s^{-1}$. The deposition velocity of 0.1 cm $s^{-1}$ accounted for the dry scanvenging of the particles. The model computes three-dimensional dispersion and depositions of the particles at downwind locations. By comparing the source strength and the downwind particle amounts, the dispersion factors (called the transfer coefficient matrix, TCM [37, 38, 41] define the transport of emissions to the downwind grids for every output time period. The TCMs calculated by the Canadian Meteorological Centre (CMC) ATDM model were used in this work [37, 38, 40]. The CMC used high-resolution winds analyzed and produced by the Japan Meteorological Agency (JMA) [32, 38, 42] with a horizontal resolution of 5 km. The emissions of $^{137}Cs$ and $^{131}I$ from 11 March to 31 March 2011 followed the estimates of Terada et al. [17, 43]. The model predicts spatial and temporal variations in airborne concentrations and deposited amounts of $^{137}Cs$ and $^{131}I$ on the surface [32, 37, 41]. These values were converted to the air dose rates [34] for comparison with the air dose rates. Meteorological analysis from the National Centers for Environmental Prediction (NCEP) was used in this work.

## Results

### Spatial distribution of the air dose rates over the North Pacific and the Northwestern Atlantic

The continuous operations of the air dose rate sensors onboard EMC ships from 19 March 2011 to 6 June 2015 collected more than 248,864 air dose rate data points at 41,477 locations (Fig 2). The 25th, 50th, 75th, and 90th percentiles of the air dose rates are 0.07±0.02, 0.079 ±0.008, 0.090±0.009, and 0.10±0.01 $\mu Sv/h$. The mean and maximum air dose rates are 0.082 ±0.008 and 1.03±0.03 $\mu Sv/h$, respectively. Based on the regressional model analysis between the air dose rates and the FDNPS emission fluxes on the Port of Tokyo calling ships (see below), the difference between the 0.0692 (25th percentile) and 0.0892 (75th percentile) $\mu Sv/h$ is 0.02 $\mu Sv/h$, which corresponds to changes of two orders of magnitude in the FDNPS emission fluxes, from $10^6$ to $10^8$ Bq/h.

Fig 3(a) shows the spatial distribution of the air dose rates measured in March 2011, the month when the FDNPS accident occurred. We note that from the measurement data (41,477 data points), we computed the air dose rates at the 25th, 50th, 75th, and 90th percentiles as 0.07, 0.08, 0.09, and 0.10 $\mu Sv/h$. Hence, 5 color intervals are used to highlight the data values with respect to the statistical values. Data in the 75th percentile are highlighted with warm colors (organe and red), while data in the 50th percentile are highlighted with cold colors (green and sky blue). March 2011 is also when the most elevated air dose rates were observed on the sailing ships (Table 2) during the entire 51 months from March 2011 to May 2015. Elevated air dose rates (above the 75th percentile, 0.09 $\mu Sv/h$) were observed in Japan and through the regions between the Aleutian low, the Alaska low, and the North Pacific subtropical high, reaching the northeastern Pacific. Elevated air dose rates were also observed between $10°N$-$30°N$ passing through the North Pacific subtropical high. We note that the month of March 2011 is a period of large-scale atmospheric depression. Two large-scale low pressure systems are located on both side of the Bering Sea: The Aleutian Low on the left, and on the right is a surface low pressure system over the off-shore of area between the Bering Sea and the western Pacific coast of Canada. One year after the FDNPS accident, in March 2012 (Fig 3(b)), elevated

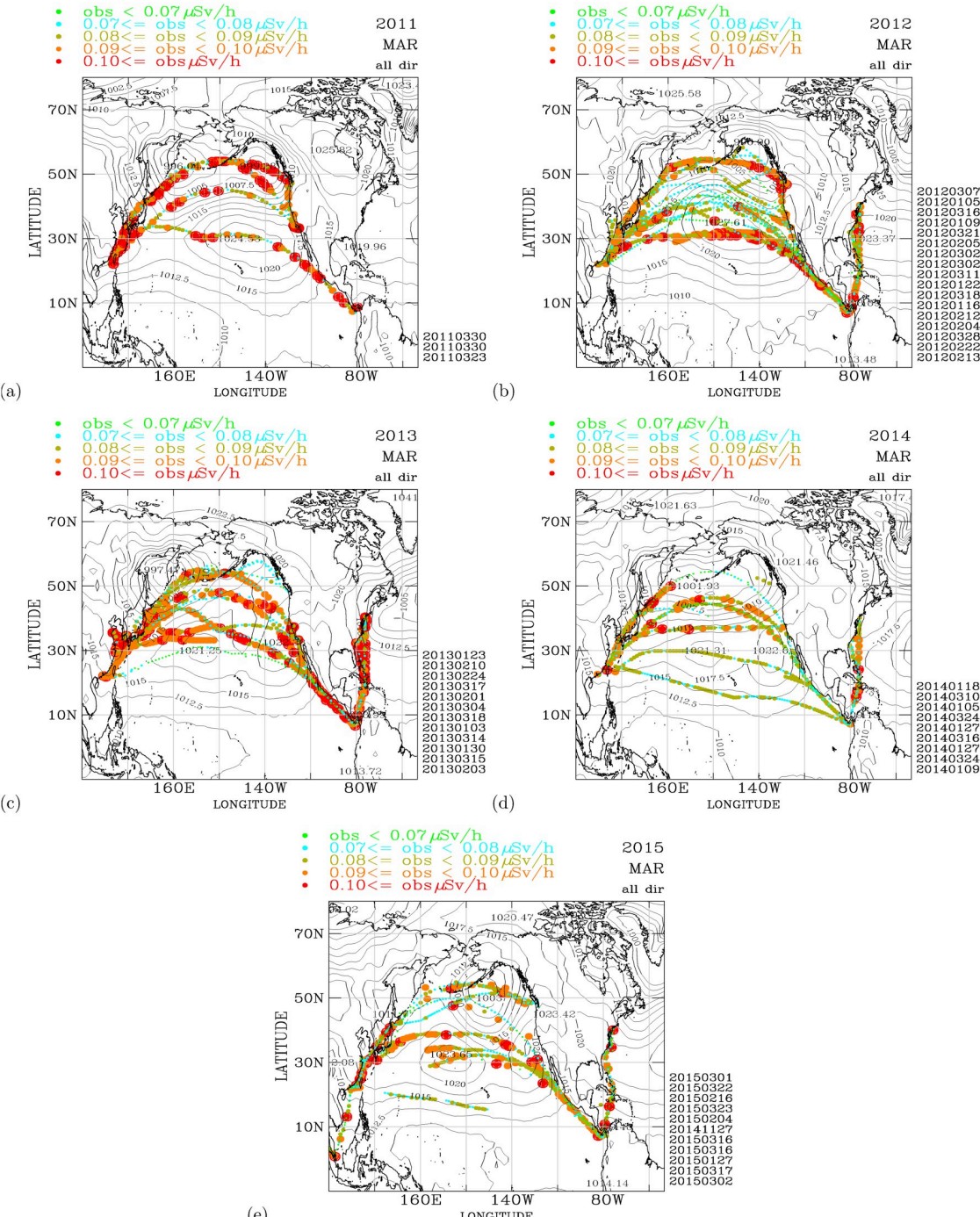

**Fig 3. Measurements of the air dose rates over the North Pacific and the northwestern Atlantic from 2011–2015.** (a) March 2011; (b) March 2012; (c) March 2013; (d) March 2014. The air dose rates are binned into five intervals: $<0.07\mu Sv/h$ (green color); $\geq 0.07$ and $<0.08\mu Sv/h$ (sky blue color); $\geq 0.08$ and $<0.09\mu Sv/h$ (dark green); $\geq 0.09$ and $<0.10\mu Sv/h$ (orange), and $\geq 0.10\mu Sv/h$ (red). The first day (e.g., 20110423 as 23 April 2011; etc.) of each cruise is shown on the bottom right of each panel. The black contours show the distribution of the monthly mean sea level pressures. Map made with Natural Earth. Free vector and raster map data at naturalearthdata.com.

**Table 2. List of monthly statistics of the 75th, and mean air dose rates (μSv/h); and the monthly FDNPS radiocesium emission fluxes (Bq/h).**

| Obs.[a] | Month[b] | Year | Month | 75th | Mean | Emissions | Rank[c] |
|---|---|---|---|---|---|---|---|
| 1 | 48 | 2015 | 2 | 0.08±0.002 | 0.07±0.002 | 0.4077E+06 | 4 |
| 2 | 47 | 2015 | 1 | 0.08±0.002 | 0.07±0.002 | 0.7800E+06 | 11 |
| 3 | 39 | 2014 | 5 | 0.09±0.003 | 0.08±0.002 | 0.5290E+06 | 7 |
| 4 | 10 | 2011 | 12 | 0.09±0.003 | 0.08±0.002 | 0.6000E+08 | 41 |
| 5 | 9 | 2011 | 11 | 0.09±0.003 | 0.08±0.002 | 0.6000E+08 | 42 |
| 6 | 38 | 2014 | 4 | 0.09±0.003 | 0.08±0.002 | 0.1050E+07 | 19 |
| 7 | 44 | 2014 | 10 | 0.09±0.003 | 0.08±0.002 | 0.5170E+06 | 5 |
| 8 | 51 | 2015 | 5 | 0.09±0.003 | 0.09±0.003 | 0.5840E+06 | 8 |
| 9 | 46 | 2014 | 12 | 0.09±0.003 | 0.07±0.002 | 0.9730E+06 | 17 |
| 10 | 49 | 2015 | 3 | 0.09±0.003 | 0.08±0.002 | 0.1160E+07 | 23 |
| 11 | 42 | 2014 | 8 | 0.09±0.003 | 0.08±0.002 | 0.8500E+06 | 14 |
| 12 | 19 | 2012 | 9 | 0.09±0.003 | 0.08±0.002 | 0.2800E+07 | 31 |
| 13 | 17 | 2012 | 7 | 0.09±0.003 | 0.08±0.002 | 0.1100E+07 | 21 |
| 14 | 50 | 2015 | 4 | 0.09±0.003 | 0.08±0.002 | 0.9520E+06 | 15 |
| 15 | 43 | 2014 | 9 | 0.09±0.003 | 0.08±0.002 | 0.3403E+06 | 2 |
| 16 | 45 | 2014 | 11 | 0.09±0.003 | 0.08±0.002 | 0.1370E+06 | 1 |
| 17 | 13 | 2012 | 3 | 0.09±0.003 | 0.08±0.002 | 0.3400E+07 | 34 |
| 18 | 15 | 2012 | 5 | 0.09±0.003 | 0.08±0.003 | 0.1100E+07 | 20 |
| 19 | 11 | 2012 | 1 | 0.09±0.003 | 0.08±0.002 | 0.7200E+08 | 43 |
| 20 | 4 | 2011 | 6 | 0.09±0.003 | 0.08±0.002 | 0.1250E+10 | 48 |
| 21 | 35 | 2014 | 1 | 0.09±0.003 | 0.08±0.002 | 0.1150E+07 | 22 |
| 22 | 8 | 2011 | 10 | 0.09±0.003 | 0.08±0.002 | 0.9000E+08 | 44 |
| 23 | 30 | 2013 | 8 | 0.09±0.003 | 0.08±0.002 | 0.3240E+07 | 33 |
| 24 | 32 | 2013 | 10 | 0.09±0.003 | 0.08±0.002 | 0.1220E+07 | 24 |
| 25 | 26 | 2013 | 4 | 0.09±0.003 | 0.08±0.003 | 0.1797E+07 | 28 |
| 26 | 22 | 2012 | 12 | 0.09±0.003 | 0.08±0.002 | 0.2900E+07 | 32 |
| 27 | 18 | 2012 | 8 | 0.09±0.003 | 0.08±0.002 | 0.8000E+06 | 12 |
| 28 | 12 | 2012 | 2 | 0.09±0.003 | 0.08±0.002 | 0.1140E+08 | 40 |
| 29 | 14 | 2012 | 4 | 0.09±0.003 | 0.08±0.002 | 0.7500E+07 | 38 |
| 30 | 6 | 2011 | 8 | 0.09±0.003 | 0.08±0.002 | 0.1999E+09 | 45 |
| 31 | 7 | 2011 | 9 | 0.09±0.003 | 0.08±0.002 | 0.2180E+09 | 46 |
| 32 | 23 | 2013 | 1 | 0.09±0.003 | 0.08±0.002 | 0.5200E+07 | 36 |
| 33 | 21 | 2012 | 11 | 0.09±0.003 | 0.08±0.002 | 0.1000E+07 | 18 |
| 34 | 16 | 2012 | 6 | 0.09±0.003 | 0.08±0.003 | 0.8500E+07 | 39 |
| 35 | 36 | 2014 | 2 | 0.09±0.003 | 0.08±0.002 | 0.1380E+07 | 26 |
| 36 | 3 | 2011 | 5 | 0.10±0.003 | 0.08±0.003 | 0.1390E+10 | 49 |
| 37 | 41 | 2014 | 7 | 0.10±0.003 | 0.09±0.003 | 0.6100E+06 | 9 |
| 38 | 40 | 2014 | 6 | 0.10±0.003 | 0.09±0.003 | 0.4140E+07 | 35 |
| 39 | 37 | 2014 | 3 | 0.10±0.003 | 0.09±0.003 | 0.1320E+07 | 25 |
| 40 | 31 | 2013 | 9 | 0.10±0.003 | 0.08±0.003 | 0.5200E+06 | 6 |
| 41 | 29 | 2013 | 7 | 0.10±0.003 | 0.08±0.003 | 0.3700E+06 | 3 |
| 42 | 28 | 2013 | 6 | 0.10±0.003 | 0.09±0.003 | 0.6300E+06 | 10 |
| 43 | 27 | 2013 | 5 | 0.10±0.003 | 0.09±0.003 | 0.8200E+06 | 13 |
| 44 | 25 | 2013 | 3 | 0.10±0.003 | 0.09±0.003 | 0.2501E+07 | 29 |
| 45 | 5 | 2011 | 7 | 0.10±0.003 | 0.08±0.002 | 0.1000E+10 | 47 |
| 46 | 24 | 2013 | 2 | 0.10±0.003 | 0.09±0.003 | 0.5800E+07 | 37 |
| 47 | 34 | 2013 | 12 | 0.10±0.003 | 0.09±0.003 | 0.9700E+06 | 16 |

(*Continued*)

**Table 2.** (*Continued*)

| Obs.[a] | Month[b] | Year | Month | 75th | Mean | Emissions | Rank[c] |
|---|---|---|---|---|---|---|---|
| 48 | 33 | 2013 | 11 | 0.10±0.003 | 0.09±0.003 | 0.2798E+07 | 30 |
| 49 | 20 | 2012 | 10 | 0.10±0.003 | 0.09±0.003 | 0.1600E+07 | 27 |
| 50 | 2 | 2011 | 4 | 0.10±0.003 | 0.09±0.003 | 0.5600E+10 | 50 |
| 51 | 1 | 2011 | 3 | 0.11±0.003 | 0.10±0.003 | 0.2690E+14 | 51 |

[a] Ranked air dose rates, from low (1) to high (51), according to the 75th air dose rates.

[b] Since March 2011.

[c] Ranked emission fluxes, from low (1) to high (51).

air dose rates similar to those that occurred in March 2011 were observed on the shipping routes. Distinctive low air dose rates (below the 50th percentile, 0.89 $\mu Sv/h$) were observed in the regions from the center of the Pacific subtropical high to approximately 50°N. The North Pacific subtropical high in March 2012 (1027.61 hPa) was stronger than that in March 2011 (1024.53 hPa), which likely helped to prevent radioactive plumes with elevated air dose rates from entering the central regions of the Pacific subtropical high. In March 2013 (Fig 3(c)), two years after the FDNPS accident, the North Pacific subtropical high was approximately 1021–1022 hPa, which was weaker than those in March 2012 and March 2011. Elevated air dose rates were observed over the shipping routes, similar to March 2011. In March 2014 (Fig 3(d)), the North Pacific subtropical high was as weak as that in March 2013, but the air dose rates were significantly lower than those in March 2013. These results are consistent with the continuing declines in the additional release of radionuclides from the FDNPS and the soil resuspension processes. In March 2015 (Fig 3(e)), four years after the FDNPS accident, occurrences of elevated air dose rates were still observed in the shipping routes. In the Supporting Information, we have included monthly distributions of the air dose rates over the North Pacific and the northwestern Atlantic in S1 Appendix of S6 Fig (March 2011-February 2012), S1 Appendix of S7 Fig (March 2012-February 2013), S1 Appendix of S8 Fig (March 2013-February 2014), S1 Appendix of S9 Fig (March 2014-February 2015), and S1 Appendix of S10 Fig (March 2015-June 2015).

Fig 3 shows the high spatial and temporal variabilities in the air dose rates over the North Pacific atmosphere observed from cruise to cruise and on a single cruise. Elevated air dose rates were observed in March 2011 and in other months during April 2011–2015. Table 2 shows the monthly statistics of the air dose rates observed on the sailing ships. The monthly statistics are ranked according to the 75th percentile of the air dose rates. Also shown in Table 2 are the FDNPS emission fluxes. Fig 4(a) shows a regression model analysis between the ranked order of the air dose rates (from low to high) versus the ranked order of the FDNPS emission fluxes (from low to high) from the data shown in Table 2.

By linking the ranks in the emission strengths to the ranks in the monthly air dose rates, the model calculated and exhibited statistically significant (P-value = 0.0298; chi square test [30] patterns between the emissions and the measurements. The regression results show that elevated air dose rates are associated with higher emission strengths. Fig 4(b) to show the trends of the monthly air dose rates over the vast eastern North Pacific, further downwind from the FDNPS. Fig 4(a) shows that the frequencies (in months) of elevated monthly air dose rates in ranks above the ranks predicted by the linear regression model mostly occurred during the summer season (June, July, August), at 7 months, followed by the spring season (March, April, May), at 6 months. The winter season (December, January, and February) and the fall season (September, October, and November) at 4 months. The spring season is active in the

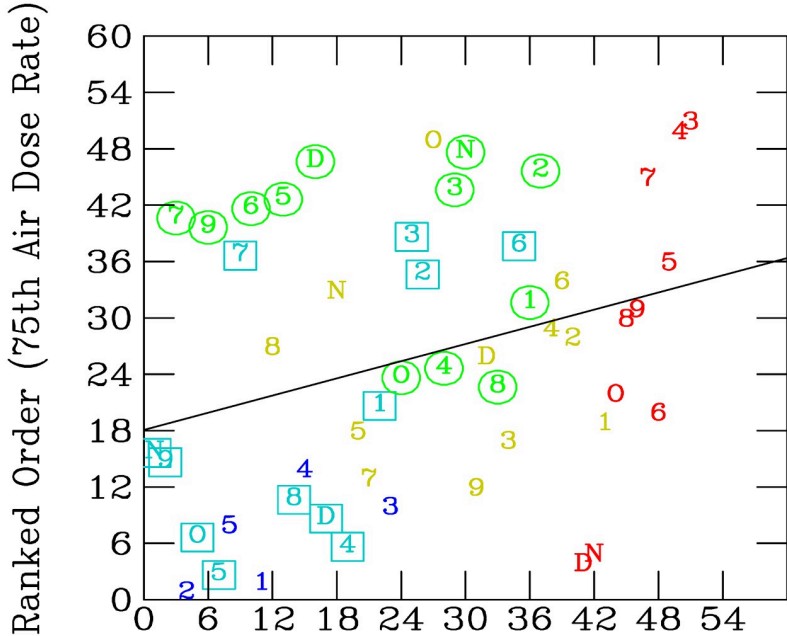

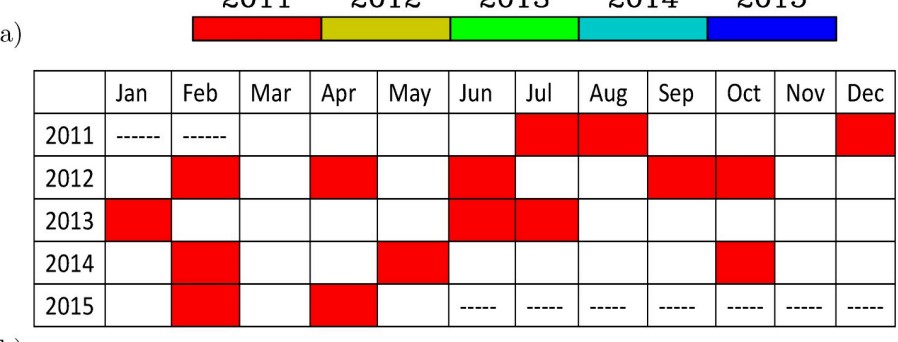

(a)

(b)

**Fig 4.** (a) The ranked order of the FDNPS emission fluxes versus the ranked order of the 75[th] percentile of the air dose rates. The measurement years are highlighted according to the color bar. The measurement months are numbered as 1 (=January), 2 (=February), etc., O (=October), N (=November), and D (=December). The line represents a linear regression analysis of the ranked order of the FDNPS emission fluxes versus the ranked order of the 75[th] percentile air dose rates. The months in 2013 are hightlighted with circles. The months in 2014 are highlithted with squares. (b) Trends of the air dose rates over the eastern Pacific from March 2011 to May 2015.

long-range transport of ground-level air pollutants from East Asia to the North Pacific. The summer season is active in the vertical pumping of ground-level pollutants by the convective atmospheric boundary layer into the troposphere. Fig 4(b) shows that March exhibits the highest frequencies of the increase in the monthly air dose rates (5 times in 5 years), followed by April, August, November, and December (4 times, respectively).

The strength of the radionuclides exported to the North Pacific atmosphere and the elevated air dose rates have been in decline. A statistically significant correlation is found between

the decrease in the elevated air dose rates and the FDNPS emission fluxes. The occurrences of the elevated air dose rates indicate the lasting impact of the FDNPS accident after March 2011. With the high density of the PGGM measurements (Fig 2), we are able to document the air dose rates over the North Pacific and the northwestern Atlantic from March 2011 to 2015 after the FDNPS accident.

## Radioactive dose rates over the land surface in the port of Tokyo

Fig 5(a) shows time-series measurements of the air dose rates in the attic of the Port of Tokyo office from March 2011 to September 2015. In the first two months after 19 March 2011, the highest individual air dose rate measured at the Port of Tokyo office was 0.50 $\mu$Sv/h. In March 2012 and one year after the disaster, the highest monthly dose rate was 0.19 $\mu$Sv/h. Two years after the disaster, the measurements indicated 0.14 $\mu$Sv/h in March 2013, 0.13 $\mu$ SV/h in the third year after the disaster (2014), and 0.12 $\mu$SV in the fourth year (2015). Given the reduction trends in the monthly maximum air dose rates, elevated dose rates still occasionally occurred in August 2012, August 2013, November 2014, and August 2015. These data exhibit the rapid impact of radioactive materials from the FDNPS after the disaster occurred and the subsequent reductions in the dose rates with time. After five and a half years, the air dose rates measured in the office attic at the Port of Tokyo still decreased. These data reveal the direct impact of the radioactive materials transported from the FDNPS to the Port of Tokyo. Fig 5(b) shows time-series measurements of the air dose rates at gate A4. The distinctive and elevated air dose rates occurred in the first year after the 11 March 2011 disaster. The air dose rates continuously decreased with a short period of elevated air dose rates that occasionally occurred from March 2011 to September 2015.

## Radioactive dose rates over the water surface at the port of Tokyo

Fig 5(c) shows time-series measurements of the air dose rates on the calling ships in the Port of Tokyo. The first year after 19 March 2011, the monthly mean air dose rates were above 0.07 $\mu$Sv/h for 11 of 12 months. In the second year (2012), the monthly mean air dose rates were above 0.07 $\mu$Sv/h for 4 of 12 months. In the third year (2013), the monthly mean air dose rates were higher than 0.07 $\mu$Sv/h for 4 of 12 months. In the fourth year (2014), the monthly mean air dose rates were higher than 0.07 $\mu$Sv/h for 2 of 12 months. In the 6 months after March of the fifth year (2015), all monthly mean air dose rates were lower than 0.07 $\mu$Sv/h. As shown before, 0.07 $\mu$Sv/h represents the 25th percentile of the air dose rates measured on all the sailing ships. Hence, the measured dose rates exhibit a gradual reduction toward the background air dose rates.

Fig 5(d) shows the time-series measurements of the air dose rates over the northwestern Pacific regions (from 120°$E$ to 180°$E$ and from 20°$N$ to 80°$N$). The maximum air dose rates close to 0.18 $\mu$Sv/h were measured during April-July 2011. The transport of radioactive materials from the FDNPS and Japan areas after 11 March 2011 was measured over the northwestern Pacific. The radiation dose rates decreased in the subsequent months of August to December 2011. The dose rates increased again from December 2011 to January 2012, during May-August and in October 2012; in January, March, and June-September 2013; and in July-October 2014. Importantly, these are also periods showing existence of a local maximum of elevated $^{137}Cs$ measurements in the surface water of the FDNPS (see Fig 10 and in Fig 2 of Aoyama et al. [9]. Hence, these data indicate that sporadic waves of radioactive materials were deposited into the water and released into the air. The measurements were able to pick up signals from these unusual events.

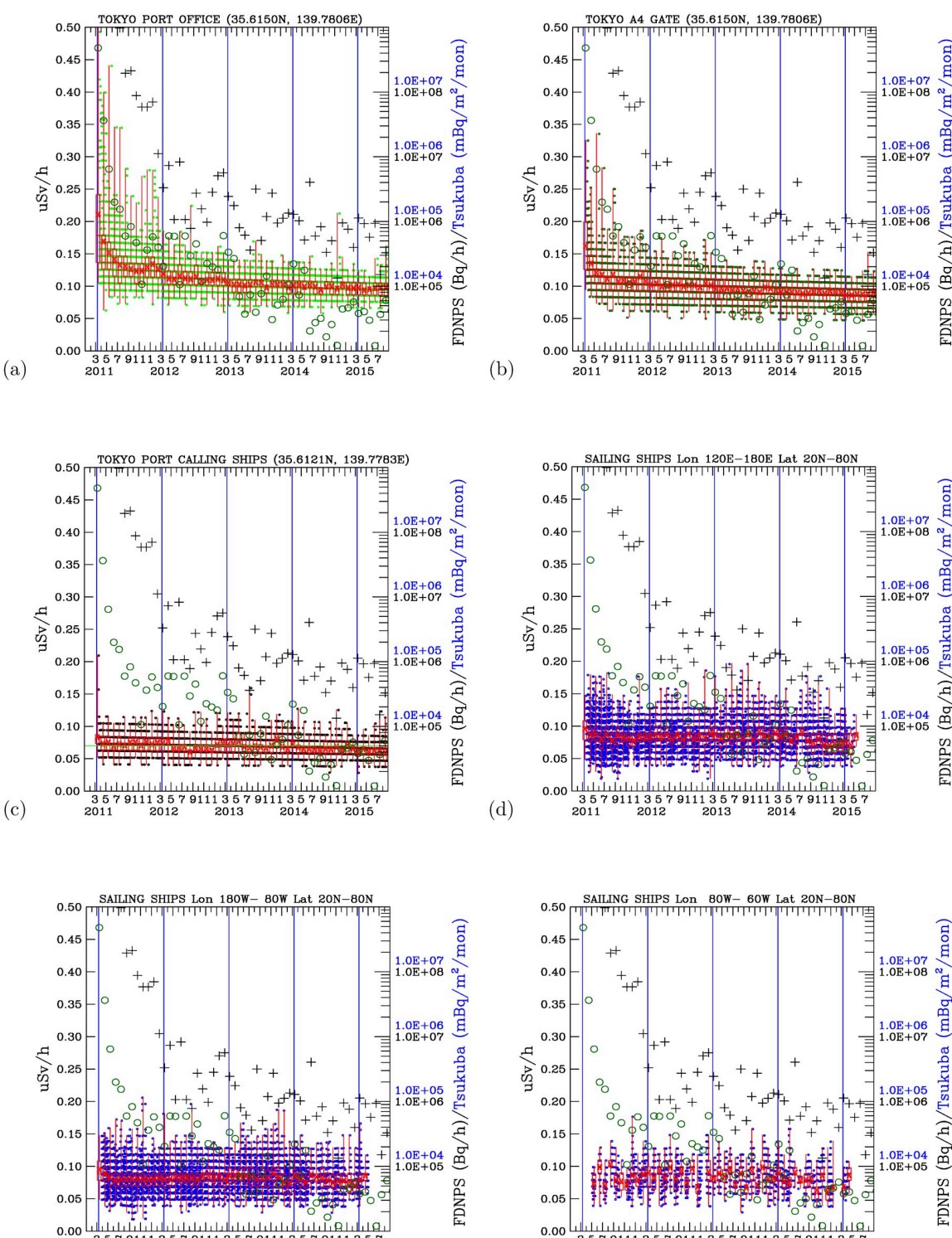

**Fig 5. Time-series radioactivity dose rates, FDNPS emission fluxes (black crosses), and Tsukuba deposition fluxes (open circles).**
Measurements over the land surface sites at the (a) Port of Tokyo office and (b)Port of Tokyo gate A4. Measurements of (c) Port of Tokyo calling ships and sailing ships over (d) the western North Pacific; (e) the eastern North Pacific; and (f) the western North Atlantic. Green dots indicate all measurements over Tokyo, while blue dots indicate all measurements on ships. The red box indicates the 25[th], 50[th], and 75[th] percentiles of all data in a month. The vertical red line indicates the maximum and minimum air dose rates (from all directions) measured in a month. The vertical blue lines indicate 11 March in 2011, 2012, 2013, 2014, and 2015.

Fig 5(e) shows time-series measurements of the air dose rates over the northeastern Pacific, further downwind of the FDNPS and Japan areas. Fig 4(b) summarizes the trends of the monthly air dose rates from Fig 5(e). Fig 4(b) shows that March exhibits the highest frequencies of the increase in the monthly air dose rates (5 times in 5 years), followed by April, August, November, and December (3–4 times, respectively).

These observed patterns of increases and decreases in the air dose rates indicate the impact of the radioactive materials being transported from land areas to the northeastern Pacific. The months of increase in the air dose rates mostly occur from November to April and from June to August. The November-April months coincide with the active long-range transport of Asian dust and pollutants from Asian land areas downwind, passing the Japan area to the northeastern Pacific. This pattern indicates that long-range transport is an important mechanism responsible for the elevated air dose rates over the northeastern Pacific. As such, ship-based air dose rate measurements confirm for the first time that the northeastern Pacific was impacted by the long-range transport of radioactive materials from the FDNPS accident.

Fig 5(f) shows time-series measurements of the air dose rates further downwind of the northeastern Pacific, from $80°W$ to $60°W$ and over the northwestern Atlantic. The observed air dose rates show increasing trends during May-June, July-August, and November-December in 2011. This is in stark contrast to the measurements made over the Pacific areas, which show decreasing trends in the air dose rates during March-April 2011 over the northwestern Pacific and during April-June 2011 over the northeastern Pacific. The direct impacts of the radioactive materials prevailed over the northwestern Pacific areas (March-April), then the northeastern Pacific (Apri-June), and finally reached the northwestern Atlantic (May-June).

## Effect of the ground level deposition of radioactive materials: The port of tokyo calling ships

Fig 6(a) compares the measurements at the two land-site locations in the Port of Tokyo: the attic of the office building and gate A4. Both measurements show consistently distinctive and elevated air dose rates in the twelve months after 11 March 2011. The high air dose rates decreased with time. However, distinctive discrepancies exist between the measurements in the attic of the office and those at gate A4. The measurements in the office attic were higher than the measurements made at gate A4. The discrepancies between these two datasets are found close to 11 March 2011 and start to converge in September 2015. The main reason for these discrepancies is the altitudes of the measurements taken in the attic and gate A4. Two additional measurements were taken in the attic of the office. The higher dose rates measured in the attic compared with those at gate A4 indicate the presence of deposited radioactive materials on the ground. The highest radioactivity levels were measured at 10 cm from the ground, followed by 50 cm and 1 m from the ground. Elevated levels were measured close to the surface, indicating the effect of radioactivity from deposited radioactive materials on the surface.

Fig 6(b) compares the measurements made in the attic of the office to those made on the compass deck of the calling ships at the Port of Tokyo. The attic measurements are clearly and distinctively higher than the measurements made on the calling ships. The measurements at the Port of Tokyo were made over the land surface, where deposited radioactive materials accumulated. The measurements on the calling ships at the Port of Tokyo were made over the ocean, where deposited radioactive materials sank in the ocean. These comparisons again demonstrate the importance of the radioactive materials deposited on the ground, which

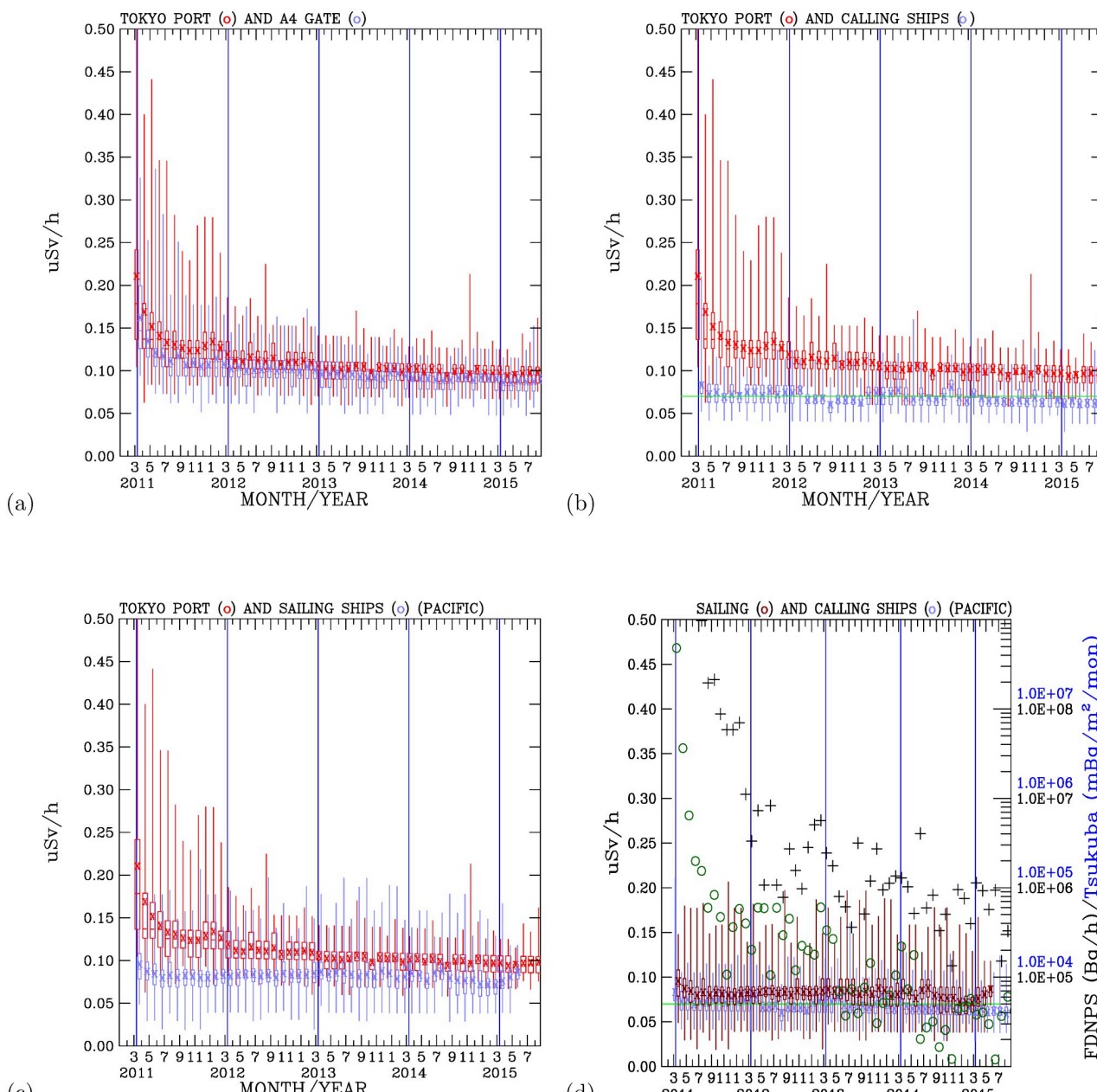

**Fig 6. Comparisons of dose rate measurements.** Measurements made at the Port of Tokyo office attic (red colored lines) compared with those made at (a) gate A4 (blue colored lines), (b) the Port of Tokyo calling ships (blue colored lines), and (c) the Pacific sailing ships (blue color lines). (d) Measurements made on the calling ships (blue colored lines) at the Port of Tokyo compared with those made on the Pacific sailing ships (dark brown lines, FDNPS emission fluxes (black crosses) and the Tsukuba deposition fluxes (blue open circles).

constituted the largest source for the measured air dose rates. On 28 August 2015, 53 months after 11 March 2011, the monthly mean air dose rate over the land-based site at the attic of the office building was 0.100±0.009 $\mu$Sv/h, compared with the 0.070±0.007 $\mu$Sv/h measured on the calling ships over the ocean water in the Port of Tokyo. The presence of radioactive materials over the land still exerted an effect on the air dose rates compared with those over the ocean.

## Effect of the ground level deposition of radioactive materials: Pacific sailing ships

To further compare measurements made over the land surface to the measurements made over the ocean surface, Fig 6(c) shows measurements made in the attic of the office building over the land surface and the measurements made on the compass deck of the Pacific sailing ships. Elevated dose rates were measured on Pacific sailing ships for several months after 11 March 2011 (more results are shown in Section 3.6). The measurements over the land surface at the Port of Tokyo were consistently higher than the measurements on the Pacific sailing ships. The differences between the land surface measurements and the Pacific measurements varied with time. Both measurements were closer to each other after 11 March 2013 than before 11 March 2013. Larger differences between the land surface and the Pacific measurements occurred close to 11 March 2011, indicating the effect of deposited ground-level radioactive materials. The gradual convergence of the air dose rates after 11 March 2013 and the variabilities in the closeness of the measurements over the land surface and the Pacific Ocean indicate that waves of radioactive materials were transported from land areas to the North Pacific atmosphere. The differences between the land-surface measurements at the Port of Tokyo and those of the Pacific sailing ships (Fig 6(c)) are smaller than the differences between the land-surface measurements and the ocean surface measurements on the calling ships at the Port of Tokyo (Fig 6(b)).

The calling ships at the Port of Tokyo include container ships traveling over the Pacific and Indian Oceans. Oceanic winds from low latitudes with low air dose rates can also impact the Port of Tokyo. On the other hand, Pacific sailing ships travel in oceanic regions downwind of Japan's land areas. Hence, the Pacific sailing ships were impacted by air plumes carrying radioactive materials from the land areas downwind to the marine environment. Interception of the radioactive materials by the Pacific sailing ships increased the dose rates measured onboard the compass desk. The Pacific sailing ship measurements are especially close to the land-based measurements during the autumn-winter-spring months, when the North Pacific is downwind of the airflow from the continent of Asia. The ships intercepted air plumes with elevated levels of radioactive materials, resulting in the enhancement of the air dose rates onboard the ships. Fig 6(d) compares the measurements on the Pacific sailing ships with those measured on the Port of Tokyo calling ships. The mean and maximum air dose rates from the Pacific sailing ships are consistently higher than those measured on the Port of Tokyo calling ships. These are important results. As the North Pacific atmosphere is directly downwind of Japan, the elevated air dose rates indicate the impact of radioactive materials being transported out of Japan toward the North Pacific atmosphere. The impacts of the elevated air dose rates were higher over the North Pacific than on the calling ships over the ocean surface at the Port of Tokyo. On the other hand, the air dose rates over the North Pacific were consistently lower than those measured over the Port of Tokyo water surface. Apparently, the cleaner air over the wide-open Pacific contributed to the Pacific measurements being lower than the Port of Tokyo measurements. The large variabilities in the air dose rates over the North Pacific indicate the North Pacific's sensitivity to upwind sources of air.

In the Supporting Information, we compare the air dose rates measured on the Pacific sailing ships that had returned to the Port of Tokyo and the measurements on the Port of Tokyo calling ships. Various instruments were used on the calling ships (Table 1) to obtain the air dose rates at the same sites in the Port of Tokyo where container ships were berthed. These comparisons aim to verify the unique measurements from the Port of Tokyo calling ships. S1 Appendix of S5 Fig compares and shows that the ranges of the air dose rates measured on the sailing ships returned to the Port of Tokyo overlapped with each other. In May and June 2011,

the measurements on the sailing ships contained large variations in the air dose rates. The measurements on the calling ships are within the ranges of the sailing ships during the two months. The measurements (25th-75th percentiles) on the sailing ships are in the upper ranges of the measurements on the calling ships in September 2011, March 2012, June 2012, August 2012, October 2012, March 2013, July 2013, and October 2014. In September 2012, the measurements on the sailing ships were within the measurements on the calling ships. In November 2014, the measurements on the sailing ships were in the lower ranges of the measurements on the calling ships. With the detailed calibrations shown in S1 Appendix of S1, S5 Figs shows that the measurements are consistent with each other.

## Associating air dose rates with emission and deposition fluxes

Figs 5 and 6(d) superimpose the monthly emission fluxes of $^{137}Cs$ and $^{134}Cs$ (in the units of Bq/h) from the FDNPS [15, 16] and the deposition fluxes (in the units of Bq/m$^2$/month) at Tsukuba [14] with the air dose rates at various measurement sites. These time-series data enable us to determine if there are associations between the air dose rates and the FDNPS emission fluxes and the Tsukuba deposition fluxes.

Fig 7(a) and 7(b) shows a scatter plot analysis of the monthly mean and the 75th-percentile air dose rates, respectively, versus the FDNPS emission fluxes and the Tsukuba deposition

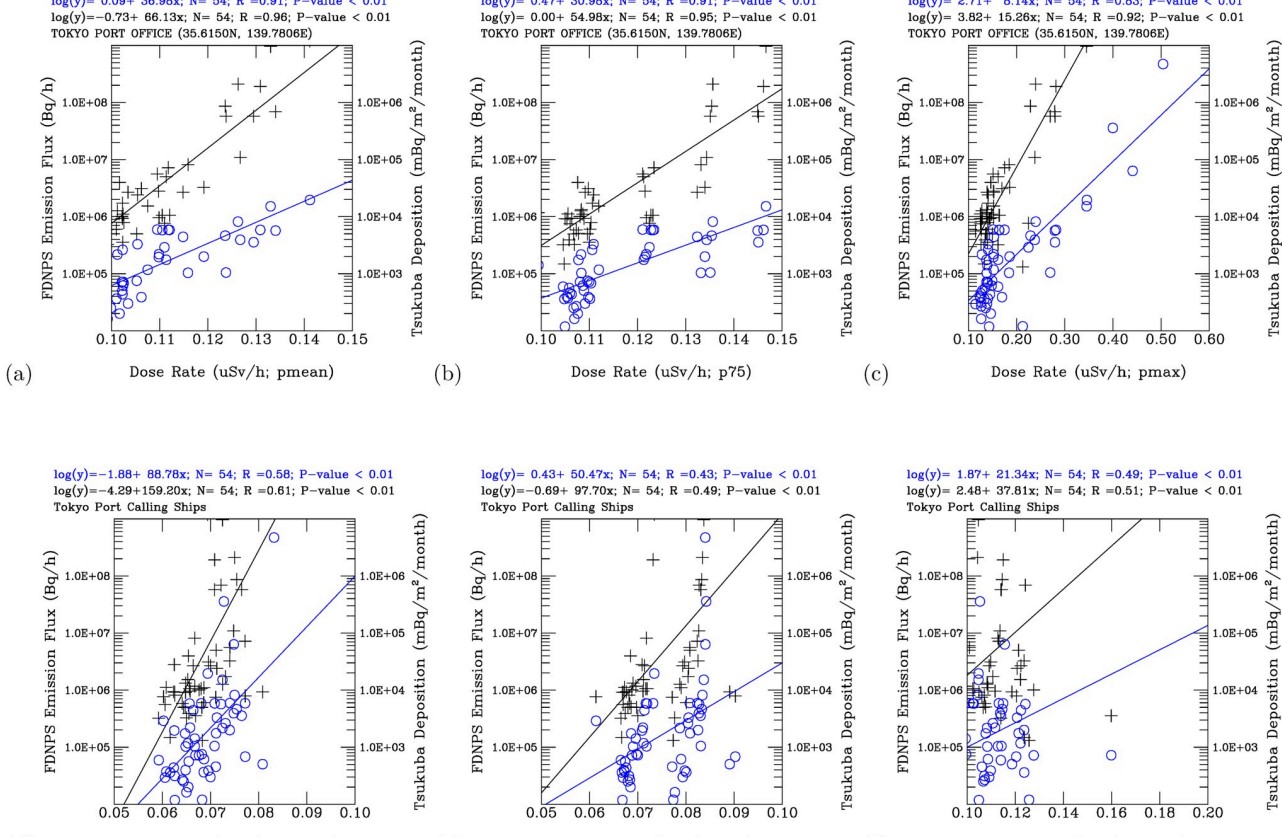

**Fig 7. Scatter plot analysis of the FDNPS radiocesium emission fluxes (black crosses) and measured Tsukuba radiocesium deposition fluxes (blue circles) versus the monthly mean (pmean), 75th percentile (p75), and maximum (pmax) air dose rates.** Port of Tokyo office: (a) pmean (b) p75 and (c) pmax. Port of Tokyo calling ships: (d) pmean, (e) p75 and (f) pmax.

fluxes at the Port of Tokyo office. The time-series data of the air dose rates are strongly ($R = 0.95$) and significantly ($P$-value $< 0.01$) correlated with the logarithm of the FDNPS emission fluxes. The air dose rates are also strong ($R = 0.91$) and significantly ($P$-value $< 0.01$) correlated with the logarithm of the Tsukuba deposition fluxes. The monthly maximum air dose rates are strongly correlated with the logarithm of the FDNPS emissions fluxes ($R = 0.92$, $P$-value $< 0.01$) and the logarithm of the Tsukuba deposition fluxes ($R = 0.83$, $P$-value $< 0.01$) (Fig 7(c)). Hence, the elevated air dose rates measured at the Port of Tokyo office are associated with the elevated emission fluxes from the FDNPS and the elevated deposition fluxes at Tsukuba. The Port of Tokyo office is located over the land surface, where both the surface-deposited and airborne radionuclides contributed to the measured air dose rates. In contrast, Fig 7(d) and 7(e) shows a scatter plot analysis of the mean and the 75th percentile air dose rates, respectively, versus the FDNPS emission fluxes over the Port of Tokyo calling ships, where the airborne radionuclides contributed to the air dose rates. Statistically significant correlations exist between the air dose rates and the logarithm of the FDNPS emission fluxes ($R = 0.61$ (mean) and $0.49$ (75th percentile), $P$-value $< 0.01$) and the logarithm of the Tsukuba deposition fluxes ($R = 0.58$ (mean) and $0.43$ (75th percentile), $P$-value $< 0.01$). Statistically significant correlations are also found between the maximum air dose rates and the logarithm of the FDNPS emissions fluxes ($R = 0.51$, $P$-value $< 0.01$) and the Tsukuba deposition fluxes ($R = 0.49$, $P$-value $< 0.01$), Fig 7(f). Hence, the analyses in Fig 7(d)–7(f) confirm that the concentrations of the airborne radionuclides contributed directly to the air dose rates measured on the calling ships over the seawater. The airborne radionuclides were the sources of the deposited radionuclides through the deposition process. As there is more variability in the amount of airborne radionuclides than in the amount of deposited radionuclides on the ground, the correlation coefficients over the land surface are higher than those over seawater.

For the measurements of the air dose rates on the sailing ships over the North Pacific Ocean, Fig 8(a) shows that the monthly 75th percentile air dose rates are significantly correlated with the logarithm of the FDNPS emission fluxes ($R = 0.39$, $P$-value $< 0.01$) and the logarithm of the Tsukuba deposition fluxes ($R = 0.36$, $P$-value $< 0.01$). Specifically, over the northwestern Pacific Ocean ($120°E − 180°E$; Fig 8(b)), the 75th percentile air dose rates are $R = 0.40$ and $P$-value $< 0.01$ correlated with the logarithm of the FDNPS emission fluxes and $R = 0.39$ and $P$-value $< 0.01$ correlated with the Tsukuba deposition fluxes. Over the northeastern Pacific Ocean ($180°W − 120°W$, as shown in Fig 8(c)), the air dose rates are $R = 0.29$ and $P$-value $= 0.0424$ correlated with the logarithm of the FDNPS emission fluxes and $R = 0.27$ and $P$-value $= 0.0568$ correlated with the logarithm of the Tsukuba deposition fluxes. Over the northwestern Atlantic Ocean ($80°W − 60°W$, as shown in Fig 8(d)), the monthly 75th percentile air dose rates are positively correlated with the logarithm of the FDNPS emission fluxes and the logarithm of the Tsukuba deposition fluxes. The scatter plot analysis of the mean air dose rates versus the FDNPS radiocesium emissions and the Tsukuba deposition fluxes are shown in S1 Appendix of S11 Fig We note that the correlation coefficients are higher and statistically more significant for the logarithm of the Tsukuba deposition fluxes ($R = 0.25$, $P$-value $= 0.0912$) than for the logarithm of the FDNPS emission fluxes ($R = 0.13$, $P$-value $= 0.4005$). This indicates that the resuspension of the deposited radionuclides [16] becomes a more dominant source than the direct emissions from the FDNPS in the long-range transport of radionuclides across the Pacific Ocean and North America. The long-range transport phenomenon of FDNPS radionuclides has been presented well in Povinec et al. [33] and resembles the long-range transport of Asian dust clouds [44]. We also note that the correlation coefficients $R$ decrease with increasing distance of the measurement sites from the FDNPS. The values vary from 0.95 (over the land surface at the Port of Tokyo office) to 0.49 (Port of Tokyo calling ships), to 0.40 (over the northwestern Pacific Ocean), to 0.29 (over the northeastern

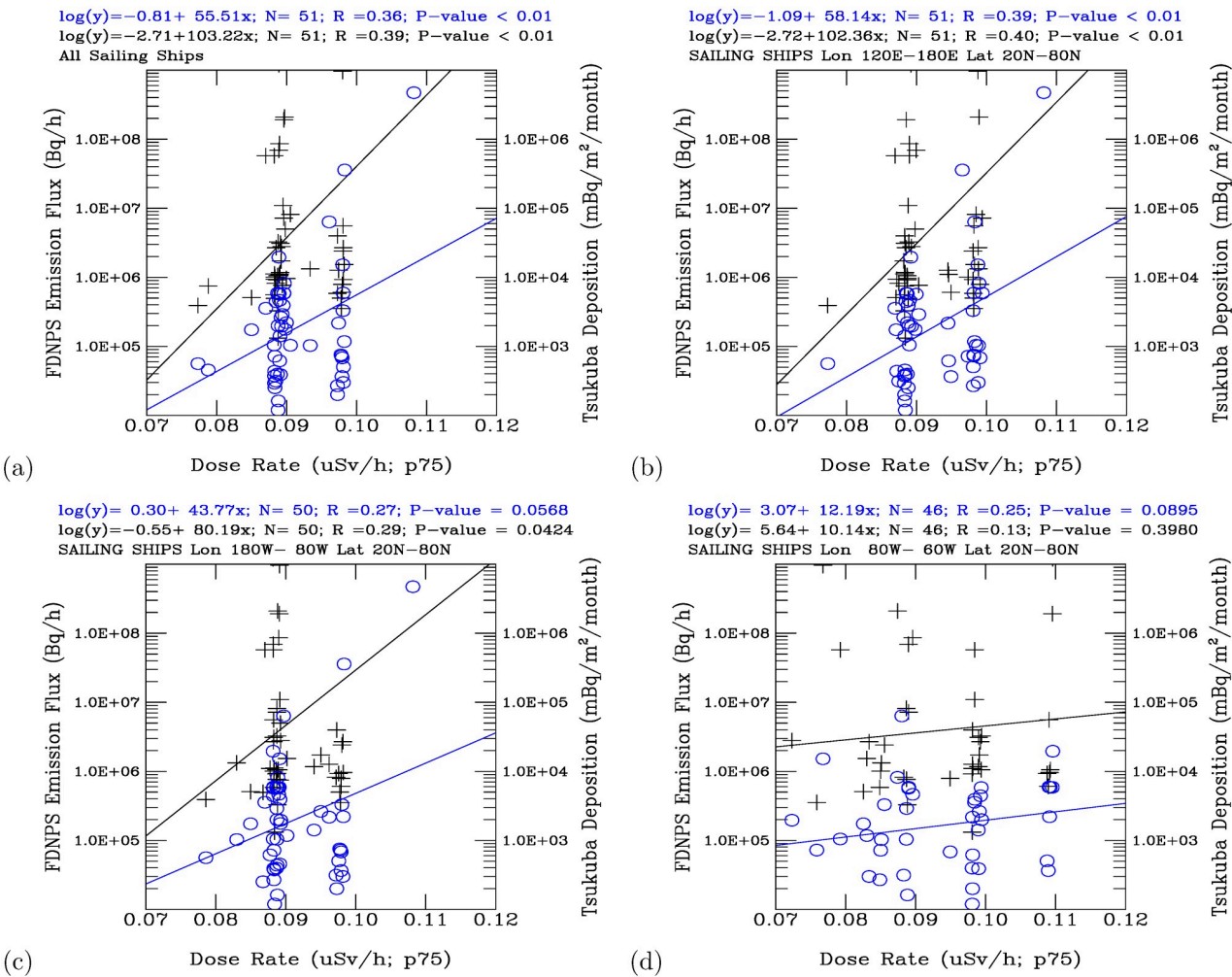

**Fig 8. Scatter plot analysis of the FDNPS radiocesium emission fluxes (black crosses) and measured Tsukuba deposition fluxes (blue circles) versus the monthly 75[th] percentile (p75) air dose rates on sailing ships.** (a) North Pacific Ocean. (b) Northwest Pacific Ocean. (c) Northeast Pacific Ocean. (d) Northwest Atlantic Ocean.

Pacific Ocean), to 0.13 (over the northwestern Atlantic Ocean). Note that the long-range transport of gaseous $^{131}I$ and other radionuclides from the FDNPS accident was detected in southern Poland on 29 March 2011 [45].

The radionuclides from the FDNPS emissions and the resuspension processes continuously dispersed as the air flows travelled across the North Pacific Ocean over North America, finally reaching the Atlantic Ocean. The signals from the FDNPS became faint, but RadEye sensors were able to measure air dose rates that were still positively correlated with the FDNPS emission fluxes and the Tsukuba deposition fluxes. In the discussion section, we have used data from Comprehensive Nuclear-Test-Ban Treaty (CTBT) Organization (CTBTO) for assessing impacts of the FDNPS. The CTBTO measurements at Reykjavik provides good data from gamma spectrometer measurements of the CTBT-relevant radionuclides [46]. Fig 9(a) shows a statically significant correlation ($R = 0.33$, $P$-value = 0.0170) between the 75[th] percentile air dose rates of the sailing ships over the North Pacific Ocean and those of the Port of Tokyo calling ships. The correlations are weaker but still positive ($R = 0.10$, $P$-value = 0.5066) between the measurements over the northwestern Atlantic Ocean and those over the Pacific Ocean

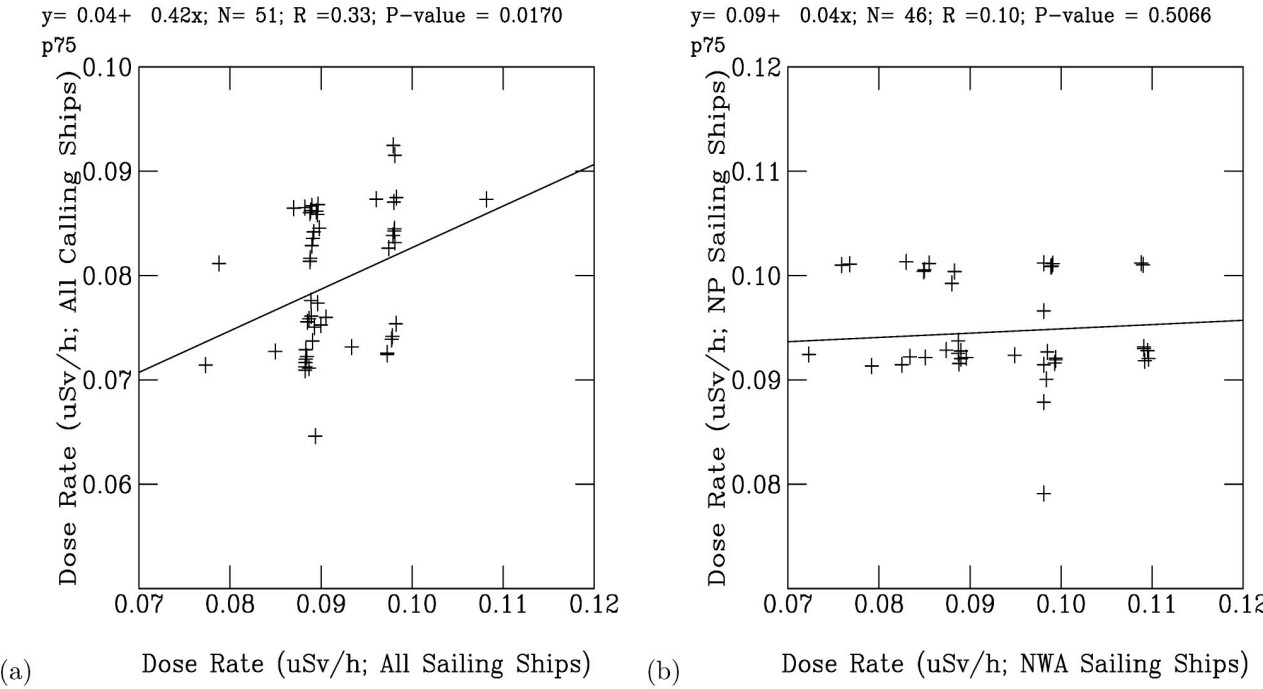

**Fig 9.** Scatter plot analysis of the monthly 75th percentile and maximum (b) air dose rates measured on the ships. (a) North Pacific Ocean sailing ships versus Port of Tokyo calling ships. (b) Northwest Atlantic sailing ships versus North Pacific sailing ships.

(Fig 9(b)). Hence, the spatial pattern of the association coefficients between the elevated air dose rates is consistent with the associations between the elevated air dose rates and the logarithm of the FDNPS emissions and the logarithm of the Tsukuba deposition fluxes. The dispersion of the positive association coefficients with increasing distance from the FDNPS highlights the long-range and long-term impacts of the FDNPS accident.

Fig 10 compares the activities of $^{137}Cs$ (on a logarithmic scale) in the surface water of canal units 5 and 6 at the FDNPS site (see Fig 2 of Aoyama et al. [9]) to the air dose rates measured in the attic of the Port of Tokyo office. The lowest activity of $^{137}Cs$ before the FDNPS accidents was approximately 1–2 Bq/m [9]. The measurements at the two canal units of the FDNPS indicate a significant input of radioactive $^{137}Cs$ into the water (from atmospheric deposition and directly discharged radiocesium from the FDNPS). The initial values were over 10 million Bq/$m^3$. It took approximately 2 years for these high levels of $^{137}Cs$ to exponentially decay to approximately thousands of Bq/$m^3$. The exponential decay of the monthly maximum levels of the air dose rates at the Port of Tokyo office remarkably resembles the pattern of the exponential decay of the logarithm of $^{137}Cs$ measured at the canal sites of the FDNPS. This comparison indicates that the FDNPS contributed to the radioactive materials in Tokyo and to the radioactive materials in the surface water of the canals of the FDNPS. The comparisons shown in Fig 10 are consistent with the findings[37] of a strong correlation between in situ measurements of $^{131}I$, $^{134}Cs$, and $^{137}Cs$ in seawater samples and gamma-ray peak counts by the aerial survey on 18 April 2011 in the FNDPS coastal area. The 75th, 50th, and 25th percentiles and the mean of the air dose rates also show decay patterns, but the reduction rates are less than those seen in the reduction rates of the monthly maximum levels of dose rates. Interestingly, the monthly lowest levels of dose rates were persistently close to 0.07 $\mu$Sv/h, except in September 2012. These lowest dose rates were not perturbed by the FDNPS accidents, indicating the background levels of the air dose rates over the Tokyo area.

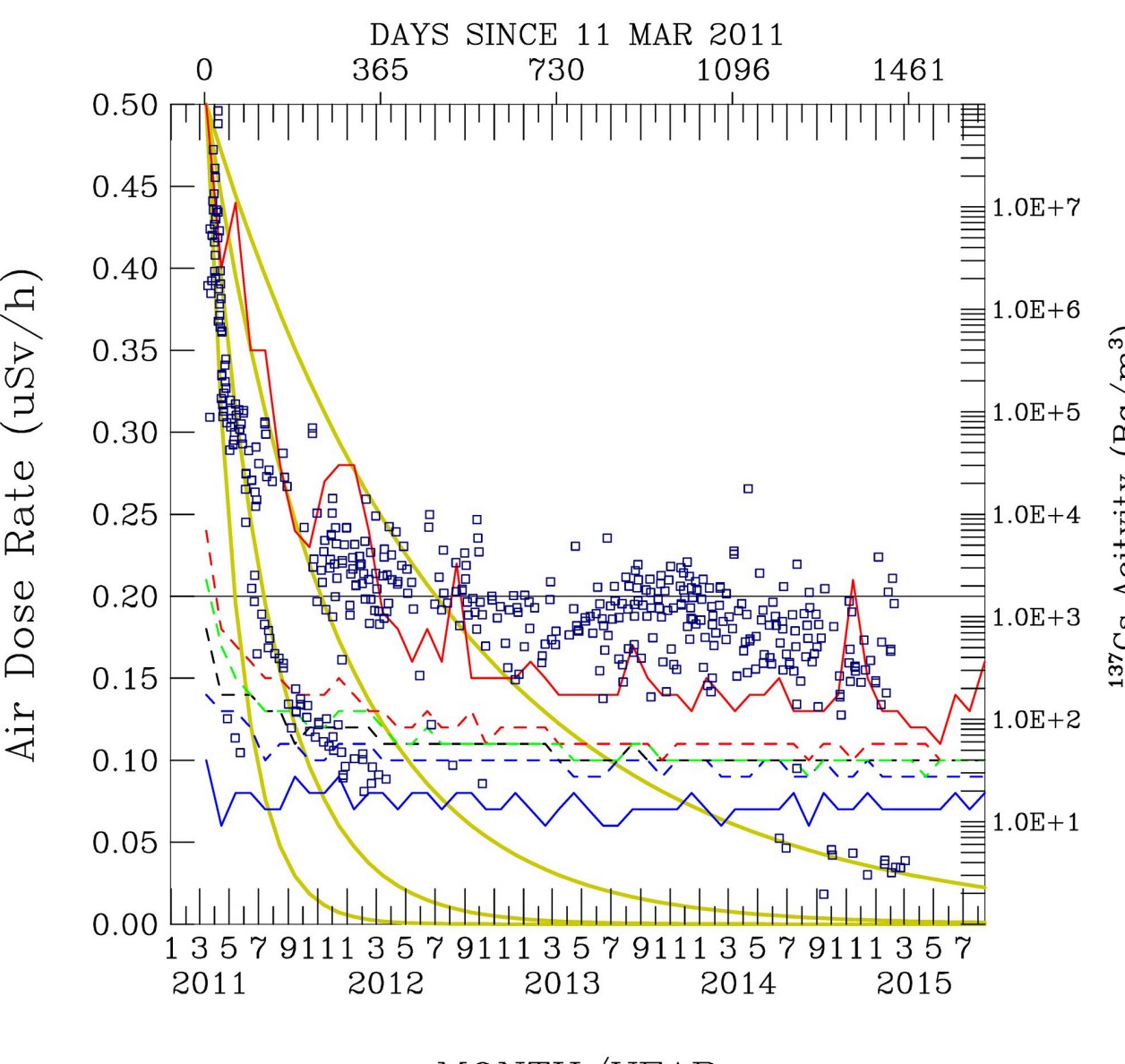

**Fig 10. Comparisons of radioactivity measured at the Port of Tokyo with the radioactive nuclei $^{137}$Cs measured (blue squares, Bq/$m^3$) in a drainage by Aoyame et al. [9].** Red solid, red dashed, green dashed, black dashed, blue dashed, and blue solid lines indicate the maximum, 75th, 50th, mean, 25th, and minimum air dose rates measured at the Port of Tokyo. For reference, the yellow curves show the analytical distribution of the time-series air dose rates calculated with half-lives of 45, 90, 180, and 360 days.

S1 Appendix of S1 Table shows a list of published measurements for the air dose rates made from sites in Japan after the FDNPS accident. Elevated air dose rates of 94 $\mu$Sv/h were measured at the Daini Nuclear Power Plant (FNPP2), 15 km south of the FDNPS on 15 March 2011. The air dose rates at the Tsukuba site, located 176 km southwest of the FDNPS, were measured at 0.2–0.4 $\mu$Sv/h during 20–21 March 2011. The measurements at the Tsukuba site are consistent with the measurements at the Port of Tokyo reported in this work.

## Verifying model simulations of the radioactive clouds from the FDNPS

An important application of the air dose data collected in this work is the verification of the FDNPS emission fluxes and the global dispersion of the radioactive plume from the FDNPS. In this section, we present this verification. S1 Appendix of S12 Fig shows the simulated dispersion of the radioactivity of $^{137}Cs$ (half-life of approximately 30 years) and $^{131}I$ (half-life of approximately 8 days) at 18 UT on each day from 11 to 30 March 2011. We note that the effect of radioactivity from the noble gas $^{133}$Xe (half-life 5.247 days) was estimated to be released from the FDNPS in an order similar to $^{131}$I during 12–31 March 201138, even higher than the $^{133}$Xe released from the 1986 Chernobyl accident [47]. The simulated concentrations of $^{137}Cs$ and $^{131I}$ are in units of Bq/m$^3$, which are converted to $\mu$Sv/h by applying conversion factors of $1.29 \times 10^{-11}$ for $^{137}Cs$ and $2.44 \times 10^{-12}$ for $^{131I}$ [34]. On 11 March (S1 Appendix of S12(a) Fig), the radionuclides released from the FDNPS were transported toward the atmosphere over the Northwest Pacific Ocean. On the second day, 12 March (S1 Appendix of S12(b) Fig), the radioactive plume was transported northeastward, toward the oceanic atmosphere and toward the inland areas north of the FDNPS. On 13 March (S1 Appendix of S12(c) Fig), the radioactive plume was transported northeastward to the oceanic atmosphere. The transport of the radioactive plumes toward the inland areas south of the FDNPS started to occur on 14 (S1 Appendix of S12(d) Fig) and 15 March (S1 Appendix of S12(e) Fig). In the following 4 days, from 16 to 19 March (S1 Appendix of S12(f)–S12(i) Fig), the radioactive plumes were transported toward the oceanic atmosphere. A southward transport of the radioactive plumes to the inland areas south of the FDNPS occurred again in the following 6 days, from 20 to 25 March (S1 Appendix of S12(j)–12(o) Fig). On 26 March (S1 Appendix of S12(p) Fig), the radioactive plume was transported to the oceanic atmosphere. The plumes then turned south again for the following 4 days from 27 to 30 March (S1 Appendix of S12(q)–12(t) Fig).

How realistic are the simulations shown in S1 Appendix of S12 Fig? With the observational air dose rate data, we are able to validate the simulations. Fig 11 compares the model simulations with the observed air dose rates at 18 UT on 21 March (Fig 11(a)–11(c)), at 06 UT on 22 March (Fig 11(d)–11(f)), and at 03 UT on 23 March (Fig 11(g)–11(i)). At 18 UT on 12 March (Fig 11(a)), the southward transport of the radioactive plume occurs over 32°$N$ −34°$N$ and 137°$E$ − 140°$E$ in the simulation. In the magnified domain (Fig 11(b)), the simulated air dose rates are 0.06–0.08 $\mu$Sv/h, consistent with the observed dose rates of 0.06–0.08 $\mu$Sv/h. On 22 March (Fig 11(d)), the model simulations show southward transport of the elevated air dose rates toward inland areas (Fig 11(e)). Large spatial gradients are calculated over the Port of Tokyo area for the air dose rates (Fig 11(f)). Two PGGM observations on this day are available to verify the model simulations. The observation and the simulation at a site further north of 35°$N$ show the air dose rates at 0.14–0.16 $\mu$Sv/h. For the observation just north of 35°$N$, The observed air dose rates are 0.08–0.10 $\mu$Sv/h, while the simulated air dose rates are underestimated at 0.04–0.06 $\mu$Sv/h. Hence, the simulated southward transport of the FDNPS emissions to the Port of Tokyo area is valid, as verified by the PGGM observations. For 23 March (Fig 11(g)), the model calculates the southward transport of the radioactive plumes over the oceanic atmosphere and close to the shore of the land areas. This is in sharp contrast to the southward and inland transport of the radioactive plumes on 22 March (Fig 11(d)). The observations made at the Port of Tokyo calling ship are 0.08–0.10 $\mu$Sv/h (Fig 11(h)), The model underestimates the air dose rates at 0.04–0.06 $\mu$Sv/h. As the spatial distribution of the simulated air dose rates of 0.08–0.10 $\mu$Sv/h is close to the observations ((Fig 11(i)), the PGGM observations verify the pattern of the large spatial gradients in the simulated air dose rates. The inland areas were indeed very lucky to escape from the direct impact of the radioactive plumes on 23 March.

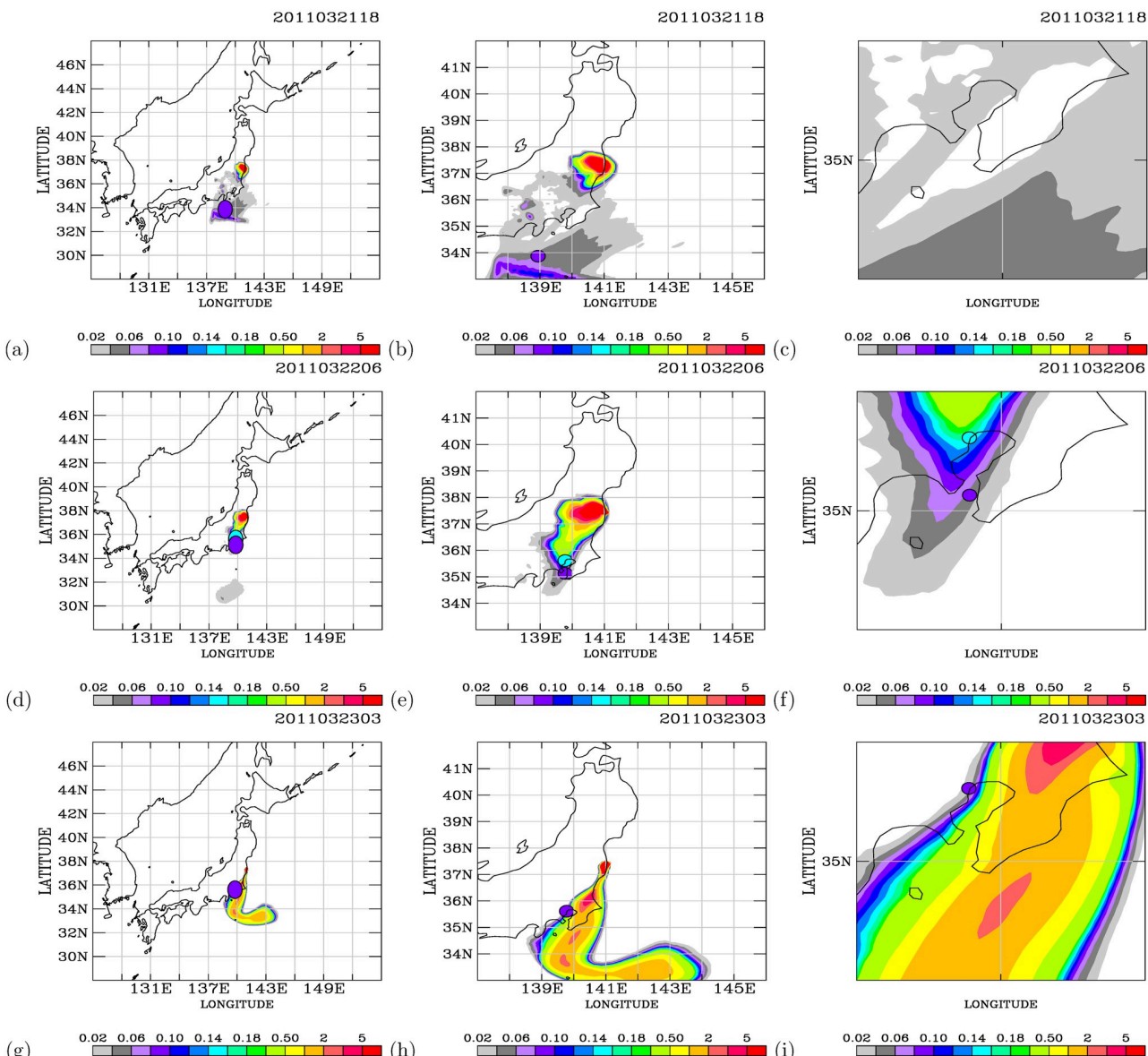

**Fig 11. Comparison of the simulated and observed air dose rates on each day for 21–23 March 2011.** Results presented in (a) large-scale, (b) magnified, and (c) Port of Tokyo area views for 18 UT on 21 March 2011. (d), (e) and (f) are the same but for 06 UT on 22 March, and (g), (h), and (i) are the same but for 03 UT on 23 March. The observed air dose rates at the corresponding times and locations are encircled with black curves. Map made with Natural Earth. Free vector and raster map data at naturalearthdata.com.

Additional comparisons between the model simulations and the PGGM observations were made at 03 UT (Fig 12(a)) and 09 UT (Fig 12(b)) on 28 March. The model calculated the southward transport of the radioactive plumes with low air dose rates ($<0.04 \mu$Sv/h). The observations show air dose rates of 0.06–0.10 $\mu$Sv/h. Hence, the background air dose rates should be approximately 0.06–0.10 $\mu$Sv/h. A distinctive southward transport of radioactive plumes in the oceanic atmosphere close to land areas was calculated on 31 March (Fig 12(c)). Large spatial gradients were calculated in the air dose rates over the oceanic atmosphere. The air dose rates observed on the Port of Tokyo calling ships were 0.06–0.08 $\mu$Sv/h. The modeled

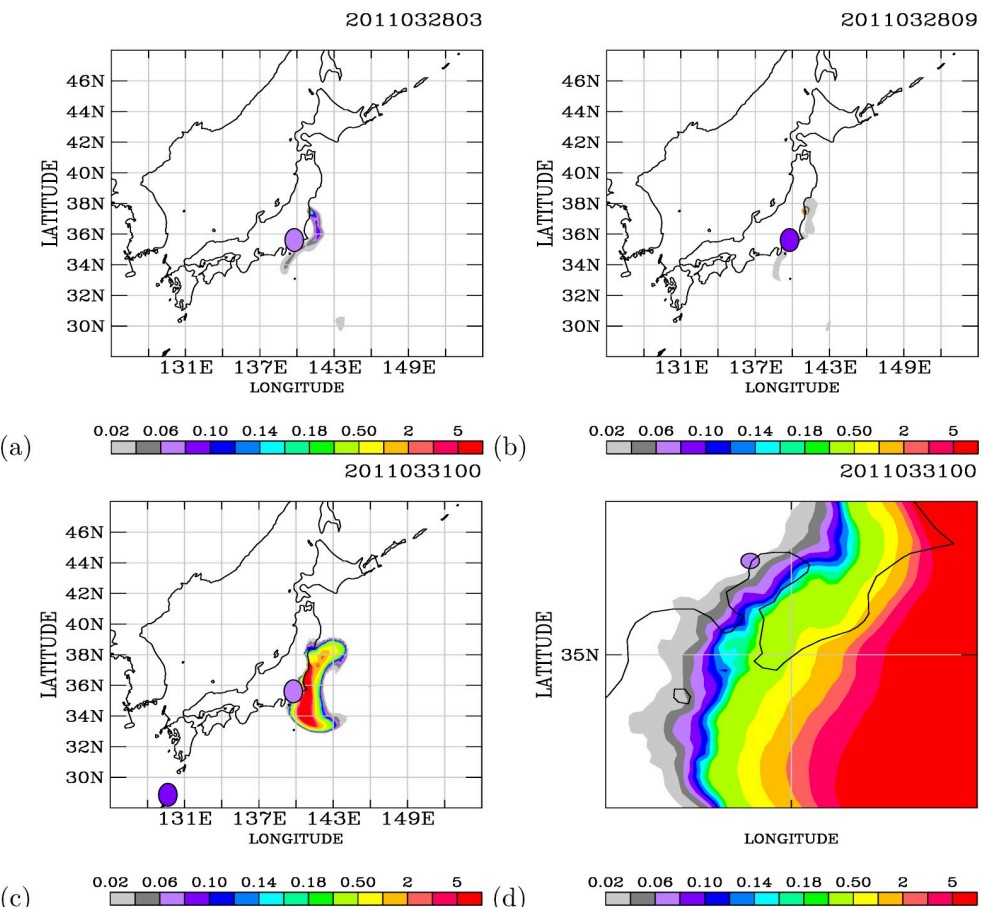

**Fig 12. The same as in Fig 11 but comparing the model simulations with the observed air dose rates at 03 UT on 28 March ((a), (b)) and at 09 UT on 28 March ((c), (d)).** Map made with Natural Earth. Free vector and raster map data at naturalearthdata.com.

air dose rates of 0.02–0.04 $\mu$Sv/h were underestimated at that exact location but were similar to the observed air dose rates in the nearby air plumes (Fig 12(d)).

The model simulations (S1 Appendix of S4 Fig, Figs 11, and 12) exhibit the spatial and temporal variability in the simulated radioactive plumes from the FDNPS. Fig 13 (a) and (b) show the time-series variations of the radionuclides $^{137}Cs$ and $^{ICs}$ (in the units of Bq/$m^3$) calculated every 3 hours over the Port of Tokyo calling ships and comparisons with the PGGM observations (in the units of $\mu$Sv/h) every 6 hours. Fig 13(a) shows that the Port of Tokyo areas were impacted by three waves of radioactive clouds from the FDNPS: during 14–15, 20–25, and 27–30 March. The PGGM observations at the Port of Tokyo calling ships started on 22 March. There are nine observations for comparison with the model results from 22 to 31 March (Fig 13(b)).

Fig 13(c) and 13(d) show comparisons of the $^{137}Cs$ and $^{131}I$ at the Tokaimura (36.4356°$N$, 140.6025°$E$) site. The Tokaimura data were taken from Draxler et al. [38]. The model shows good comparisons with the measurements, except for some periods during 16–18, 22–24, and 25–30 March 2011, when the model simulations were underestimated compared with the measurements. The underestimated simulations of $^{137}Cs$ and $^{131}I$ during 25–30 March are consistent with the underestimated simulations of the air dose rates over the Port of Tokyo during 27–30 March 2011. The underestimated simulations that occurred on 22 and 24 March are

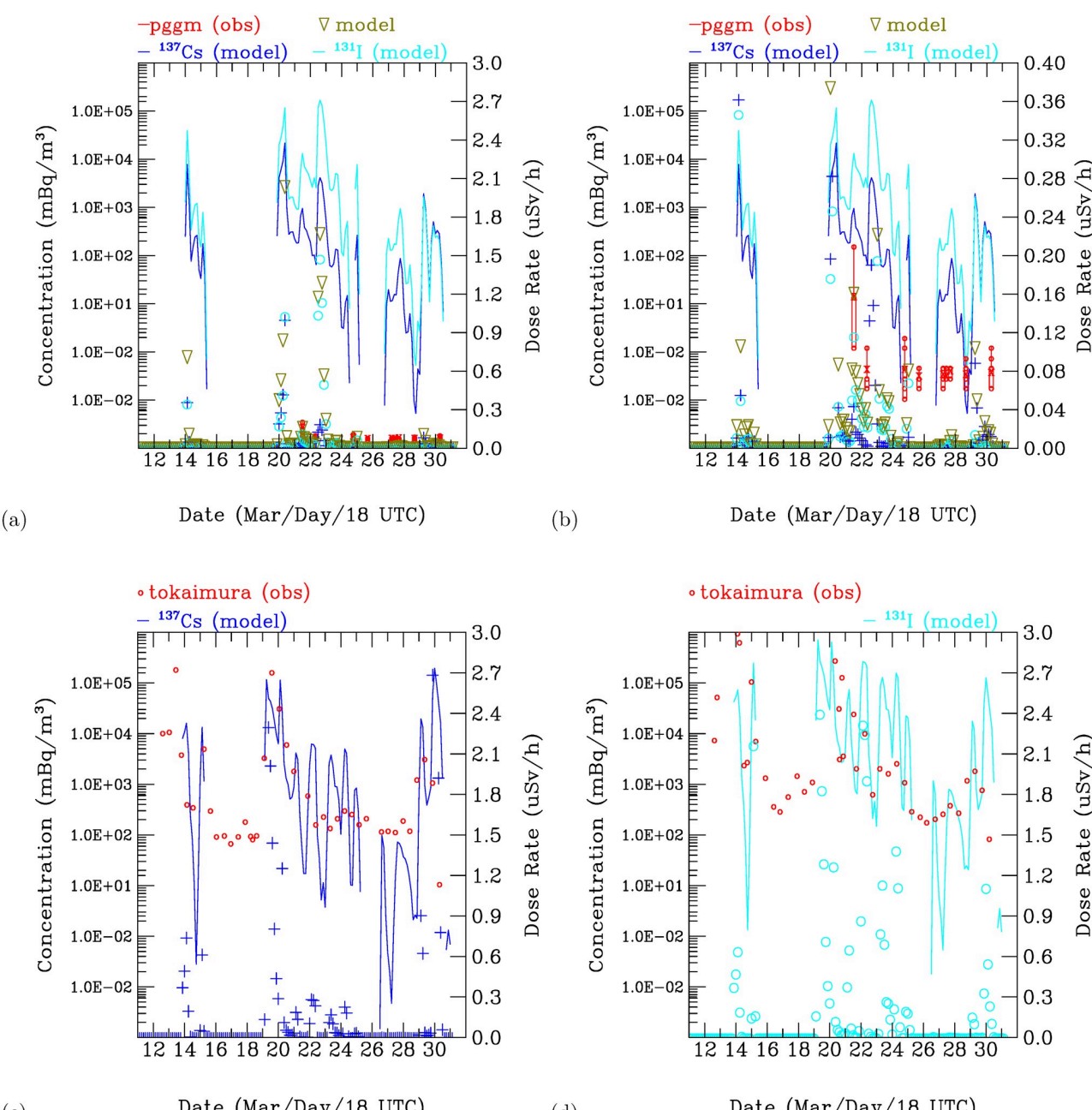

**Fig 13. Time-series variations in calculated radionuclides $^{137}Cs$ and $^{131}I$ over the Port of Tokyo area and comparisons with the PGGM observations.** The concentrations of $^{137}Cs$ (blue) and $^{131}I$ (sky blue) are in units of Bq/$m^3$ (left y-axis), the blue circles are $^{137}Cs$, the blue crosses are $^{131}I$, and the dark green triangles are $^{137}Cs+^{131}I$ in the converted units of $\mu$Sv/h. The red circles are the PGGM observations. The air dose rates range between (a) 0.0 and 3.0 and (b) 0.0 and 0.4 $\mu$Sv/h. (c) and (d) show comparisons of $^{137}Cs$ and $^{131}I$ at the Tokaimura (36.4356°N, 140.6025°E) site.

mostly due to the underestimation of $^{137}Cs$. This indicates that the underestimated air dose rates at the Port of Tokyo during 22–25 March are due to the underestimated simulations of $^{137}Cs$.

S1 Appendix of S2 Table summarizes the comparisons shown in Fig 13. At 06 UT 22 March, the simulated air dose rate is 0.16 $\mu$Sv/h, while the observed air dose rates range from

0.11–0.21 $\mu$Sv/h with mean value of 0.16 $\mu$Sv/h. Hence, the model agrees well with the observations. At 02 UT on 23 March, the simulated air dose rate is 0.03 $\mu$Sv/h, which is lower than the observed value range of 0.06–0.10 $\mu$Sv/h. There are no simulated air dose rates at 12 UT on 25 March, at 10 UT on 26 March, or at 23 UT on 27 March. The observed mean air dose rates are 0.07, 0.08, and 0.07 $\mu$Sv/h, respectively. The simulated air dose rates are underestimated compared with the observed values. The range of the simulated air dose rates at 04 UT and 10 UT on 28 March, at 10 UT on 29 March, and at 01 UT on 31 March is 0.00–0.01 $\mu$Sv/h. The mean observed air dose rates at the above times range between 0.07 and 0.08 $\mu$Sv/h. As shown in Fig 13, the radioactive plumes from the FDNPS reach the Port of Tokyo calling ship during the periods 22–25 and 27–30 March, but the model underestimates the air dose rates. Hence, the estimated emissions from the FDNPS are likely too low during these two periods. The modeled radioactive plumes do not appear between 25th and 27th, but the observed plumes show indicate air dose rates of approximately 0.08 $\mu$Sv/h. The resuspension of deposited radionuclides likely contributed to the observed air doses [16], and the model used in this work does not include the resuspension process. Additionally, the bias of the meteorology used in the ATDM model can contribute to the deviation in the transport of the radioactive plumes from the FDNPS.

## Discussion

Naturally occurring cosmic radiation accounts for 0.034 $\mu Sv/h$ (0.3 mSv/year; Akerblom et al., 2000). Over the land, the average air dose rates ranged from 0.03 (Iceland), to 0.07 (Finland), to 0.08 $\mu Sv/h$ (Denmark, Norway, Sweden) [48]. The mean air dose rate in Japan before the FDNPS accident was less than 0.0434 $\mu Sv/h$ (0.38 mSv/year [49]). We note that the air dose rates measured in the Tokyo Shinjuku (35.7065˚N, 139.6979˚E) area from the Japan Ministry of Education, Culture, Sports, Science and Technology (MEXT [50]) were used to confirm the background levels of the air dose rates over the Tokyo area. S1 Appendix of S2 Fig shows time-series measurements of the air dose rates. Before the arrival of the radioactive plume from the FDNPS accident, before 18 UT-19 UT on 14 March 2011, the air dose rates were measured at 0.033–0.035 uSv/h. After 18 UT-19 UT on 14 March 2011, the air dose rates were measured at levels higher than 0.04 uSv for the entire 2011 period. S1 Appendix of S3 Fig shows time-series modeled $^{137}$Cs and $^{131}I$ concentrations (mBq/m$^3$) and in the converted air dose rates ($\mu$Sv/h) and comparisons with the Tokyo Shinjuku measurements. The comparison shows that the first arrival of $^{137}$Cs and $^{131}$I I at Tokyo Shinjuku occurred at 18 UT on 14 March 2011. S1 Appendix of S4 Fig shows the temporal and spatial dispersion of the $^{137}$Cs and $^{131}$I radioactive clouds from the FDNPS. The arrival of the radioactive clouds occurred at 18 UT on 14 March 2011. These results show that the background level of the air dose rates around the land area of the Port of Tokyo before 18 UT on 14 March 2011 was close to 0.04 $\mu$Sv/h. Based on this reconfirmation of the air dose rates, the impact of the FDNPS accident remained in 2015. The 50th percentile of the air dose rates of 0.079 $\mu Sv/h$ over the oceanic atmosphere approximates the contributions from cosmic radiation and the terrestrial radiation associated with the transport process from Japan: 0.077 $\mu Sv/h$ (=0.034 $\mu Sv/h$ plus 0.043 $\mu Sv/h$).

In 2011 alone, March (the 1st month, bottom row of Table 2) shows the highest dose rates and emission fluxes, while December (the 10th month, top 4 row of Table 2) shows the lowest dose rates and emissions fluxes. When all 51 months of data are compared, March 2011 shows the highest emission fluxes and air dose rates, followed by April 2011 (the 2nd month after the 11 March accident). See also the statistics shown in the bottom two rows of Table 2. Except for the first two months after the FNDPS accident in 2011, the 75th percentile of the dose rates from months in 2012 and 2013 that occurred are higher than the end month of 2011. As

shown in Kajino et al. [16], resuspension of radionuclides from ground soil and forest systems contributed to the long-term assessment of airborne radiocesium after the FDNPS accident. Our data shown in this work concur with the findings of Kajino et al. [16].

The comparisons shown in Fig 10 are consistent with the findings [51] of a strong correlation between in situ measurements of $^{131}$I, $^{134}$Cs, and $^{137}$Cs in seawater samples and gamma-ray peak counts by the aerial survey on 18 April 2011 in the FNDPS coastal area. We note that the effect of radioactivity from the noble gas $^{133}$Xe (half-life 5.247 days) was estimated to be released from the FDNPS in an order similar to $^{131}$I during 12–31 March 2011 [52], even higher than the $^{133}$Xe released from the 1986 Chernobyl accident [47]. To test whether the impacts of the FDNPS were observed in the air dose rates, we used a linear regression model to find associations between the FDNPS emission fluxes and the Tsukuba deposition fluxes. We identified three key aspects of the impacts of radioactive materials following the FDNPS accident from time-series measurements and linear regression model analysis. Comparisons of the air dose rates measured at various altitudes from the ground indicate an altitude effect from transported and deposited radioactive materials on the ground.

We also used the satellite-observed aerosol distribution to compare and determine the aerosol distribution and hence the air dose rates over the North Pacific and the Port of Tokyo. Satellite data provide consistent and long-term measurements on a global scale. The absorbing aerosol index (AAI) was provided by the Ozone Monitoring Instrument (OMI) onboard the National Aeronautics and Space Administration (NASA) Aura satellite. The monthly AAI data on a 1-degree longitude-latitude system from February 2011 to December 2015 were obtained from http://www.temis.nl/airpollution/absaai.

S1 Appendix of S13 Fig shows the spatial distribution of the scaled AAI from the OMI satellite. The AAI data are presented on a horizontal resolution of a 1 degree longitude-latitude grid system. The AAI values on each grid represent the summation of the monthly AAI from February 2011 to December 2015. For comparison purposes, the original AAI was scaled by the AAI values on a grid covering the Port of Tokyo area. S1 Appendix of S13 Fig shows that the scaled AAI values over the North Pacific are 1.01 to 1.40 times those over the Port of Tokyo. Hence, aerosol loadings are more abundant over the North Pacific downstream of Japan than over the Port of Tokyo. S1 Appendix of S14 Fig shows a zoom-in analysis of the scaled AAI over the Japan region. The patterns of the higher AAI over the North Pacific than over the Port of Tokyo area are consistent with the higher air dose rate measurements over the North Pacific than over the Port of Tokyo calling ships. These results indicate the important contribution of the elevated aerosols and their associated radionuclides transported over the North Pacific than over the Port of Tokyo calling ships after the FDNPS accident.

In addition, we have used the air dose rates in the middle of the North Pacific at Hawaii (S1 Appendix of S15 Fig) to determine the background level of the dose rate impact after the FDNPS accident. The air dose rates and the gamma gross counts per minute are provided by the United States (US) Environmental Protection Administration (EPA). The data were obtained from https://www.epa.gov/radnet/radnet-near-real-time-air-data-honolulu-hi. S1 Appendix of S15 Fig shows hourly air dose rates observed at Hawaii from 2017 to 2020. No air dose rate measurements were made before 2017. The observed air dose rates are very stable. The average air dose rates are 0.032 $\mu Sv/h$. These values are very close to 0.033–0.035 $\mu Sv/h$ measured at Tokyo Shinjuku before the arrival of the FDNPS to Tokyo (see S1 Appendix of S2 Fig). Although there were no measurements of the air dose rates before 2017, there were long-term continuous measurements of the gamma gross counts per minute at Hawaii. S1 Appendix of S16 Fig shows the time-series measurement of the gamma gross counts per minute from 2010 to 2020. The data were measured from gamma channel 02 (500–1000 counts per minute; (a)), 03 (380–840 counts per minute; (b)), 04 (150–300 counts per minute; (c)), 05 (75–135

counts per minute; (d)), 06 (50–80 counts per minute; (e)), 07 (70–100 counts per minute; (f)), 08 (40–70 counts per minute; (g)), and 09 (30–50 counts per minute; (h)). The gamma counts from elevated values of (a), (b), (c), (d), and (e) clearly show that higher gamma gross counts occurred in 2011 than during the 2012–2015 period. The lower gamma gross counts in (f), (g), and (h) show no significant high values in 2011. In addition, the data also show the occurrence of short periods of elevated gamma gross counts after the FDNPS accident. S1 Appendix of S17 Fig shows the scaled gamma gross count rates per minute at Hawaii observed from 2010 to 2020. Results are scaled by the 2011 average values. The lowest scaled values are close to 60% in 2017 compared with the high gamma gross counts of (a), (b), and (c) in 2011. Additionally, the elevated scaled AAI values over the North Pacific in latitudes north and south of Hawaii are approximately 1.4 (S1 Appendix of S13 Fig). The 50$^{th}$ percentile of the air dose rates observed by the PGGM ships are 0.080 $\mu Sv/h$. Therefore, approximately (0.080 × 0.60)/ 1.4 gives air dose rates of 0.034 $\mu Sv/h$, which are close to the Hawaii mean air dose rates during 2017–2020. Hence, given aerosol loading (similar to the AAI over the Tokyo and Hawaii regions), we estimate that the background air dose rate over the North Pacific is 0.034 $\mu Sv/h$. This estimate is similar to the air dose rates over Tokyo Shinjuku before the FDNPS accident.

S1 Appendix of S18 Fig shows the time-series measurements of the air dose rates (in dark brown color) on the sailing ships in a region containing Hawaii (between $170°W$ and $140°W$, and between longitudes $10°N$ and $30°N$; see also Fig 3). The air dose rates measured on the calling ships of the Tokyo Port are shown in sky blue color for comparison. The lowest horizontal green line shows the 2017–2020 average air dose rates of 0.034 $\mu Sv/h$ measured in Hawaii. After correcting for the latitudinal (spatial) effects of 1.4, the second horizontal green line (from the bottom) shows the estimated air dose rates of 0.048 $\mu Sv/h$ during 2011–2015 (0.034 * 1.4 = 0.048) in Hawaii. After correcting for the time (temporal) effects of 1/0.6, the third horizontal green line (from the bottom) shows the estimated air dose rates of 0.057 $\mu Sv/h$ during 2011–2015 in Hawaii (0.034*1/0.60 = 0.057). After correcting for both the time and the latitudinal effects, the top horizontal green lines shows the estimated air dose rates of 0.080 $\mu Sv/h$ during 2011–2015 (0.034 * 1.4 *1/0.60 = 0.080) in Hawaii. We note that 0.080 $\mu Sv/h$ are close to the mean of the all PGGM measurements of 0.082 $\mu Sv/h$ over the north Pacific.

The comparisons show that the mean of the PGGM measurements in December 2014 and January 2015 are close to 0.048 $\mu Sv/h$, and the minimum of the PGGM measurements are close to 0.034 $\mu Sv/h$. In contrast, in April 2011, the PGGM measurements are between 0.048 and 0.080 $\mu Sv/h$. Variations in the PGGM measurements had occurred in the subsequent months before December 2015. The mean of the PGGM measurements vary about 0.080 $\mu Sv/h$, but the minimum of the PGGM measurements have gradually decreased to 0.048 $\mu Sv/h$. We note that the elevated PGGM measurements have occurred in a few months when the FDNPS emission fluxes and the Tsukuba deposition fluxes are also showing upward trends (for example, during August 2012-April 2013; during December 2013-May 2014; and in March 2015). The measurements of the air dose rates (mean, the 25° percentile, and the minimum) on the call ships of the Tokyo Port have also gradually decreased from the air dose rates higher than 0.080 $\mu Sv/h$, 0.057 $\mu Sv/h$, and 0.048 $\mu Sv/h$ in March 2011 to the air dose rates close to 0.057 $\mu Sv/h$, 0.048 $\mu Sv/h$ and 0.034 $\mu Sv/h$ in August 2015. Hence, the PGGM measurements in the oceanic areas close to Hawaii and on the calling ships of the Tokyo Port show consistent patterns of the time-series measurements of the air dose rates. The impact of the FDNPS emissions have gradually reduced and the air dose rates have gradually decreased and converged to the 0.034 $\mu Sv/h$ of the air does rates measured in Hawaii during 2017–2020.

The regression analysis of the air does rates measurements over the eastern North Pacific versus the FDNPS emission fluxes, Fig 7(c), shows a regression equation of log(y) = -0.55 + 0.89x (P-value = 0.0424), here y represents the FDNPS emission fluxes in Bq/h, and x

represents the air dose rates in $\mu Sv/h$. From this equation, 0.034 $\mu Sv/h$ corresponds to 150 Bq/h of emission fluxes. The 0.048 $\mu Sv/h$ air dose rates correspond to 1991 Bq/h of the emissions fluxes, about 13 times of the emissions fluxes to the background air dose rates of 0.034 $\mu Sv/h$. The 0.057 $\mu Sv/h$ air dose rates correspond to 10,491 Bq/h of the emission fluxes, about 70 times of the emission fluxes to the background air dose rates. The 0.080 $\mu Sv/h$ air dose rates correspond to 733,162 Bq/h of the emission fluxes, about 4,900 times of the emission fluxes of the background air dose rates.

Thanks to the full collaboration from the EMC, we were furtunately able to persistently take measurements from 2011 to 2015 at the Port of Tokyo office, gate A4, the calling ships with the same G01 instrument and on the sailing ships with the instruments from G02 to G14. With the air dose rates reported in this work, we used the ATDM model to test and verify the amount of radioactive materials emitted into the atmosphere following the FDNPS accident during 11–31 March 2011 [15, 16, 53]. We note that there were post-accident releases of radionuclides into the atmosphere or coastal waters and deposition during this period [17, 32, 43]. Comparisons showed that the model underestimated the air dose rates during the periods of 22–25 March and 27–30 March, indicating that the estimated emissions from the FDNPS were likely too low during these two periods. The underestimation of the modeled air dose rates that occurred on 22 March is consistent with the underestimation of the atmospheric $^{137}Cs$ on 22 March by recent aerosol modeling of the atmospheric dispersion of $^{137}Cs$ on 22 March (the N-model [54]). Additionally, no radioactive plumes were present in the locations where background air dose rates of 0.077–0.082 were measured. The nonexistence of the modeled radioactive plumes reflects the model's lack of mechanisms such as the resuspension of radionuclides from the ground [16] and/or the bias in the meteorology used in the ATDM calculations. The very important radionuclide resuspension process described in Kajino et al. [16] may need to be taken into account when the atmospheric inverse models [55] are used for effective source attributions from the observations.

The air dose meters (Geiger-Mueller (GM) meter) RadEye B20 and G10 were used in this work. Both GM meters have been extensively used in published works related to the measurements of air dose rates in places associated with tests of nuclear weapons, accidents from nuclear power plants, medical imaging processes with patients injected with radioactive materials, robotic exploration of radioactive environment, radioactivity safety training, etc. There are over 10 different models and more than 20 variants of the RadEye hand held radiation meters announced since 2002. The core design are the same for all models. There were more than 75,000 units in use world-wide in 2019 [24, 56, 57].

For the radiation measurements in ports, a guideline has been issued by the Ministry of Land, Infrastructure, Transport and Tourism (MLIT) of Japan [58]. The equipments to be used for the measurements of the radiation dose rate for ships in ports are air dose GM meter, or scintillator, or ionization chamber, or semi-conductor survey meter. The $\gamma$ ray is to be detected. The range for detection is from 60 kV to 1.25 MeV for $\gamma$ ray detection. Proper calibration should be confirmed. Accuracy of the measurments are ±20% for $^{137}Cs$ [58]. The RadEye B20 used in this work detects $\gamma$ ray, with a range for detection between 17 keV and 1.30 MeV, and accuracy of ±3% for $^{137}Cs$ [22]. The RadEye G10 air dose meter detects $\gamma$ ray, with a detection range from 50 keV to 1.3 MeV, and accuracy of ±3% for $^{137}Cs$ [25]. We had rigourously calibrated all sensors used in this work (as reported in the section Method). Our sensors meet the requirements issued by the MLIT.

Measurements of $^{137}Cs$ concentrations on the surface of the tree bark from four forests in Fukushima Prefecture after the Fukushima Daiichi accident. There were 48 cedar bark samples collected and measured (in the field and indoor, respectively) in 4 forests during July-September 2017 in Fukushima prefecture. Ambient air dose rates of 0.3–1.5 $\mu Sv/h$ [59] were measured

in the forests. The samples measured by the RadEye B20 (in the units of count per minute) in the field were again measured indoor with the germanium semiconductor detector GC2518 (in the units of Bq/kg; [60, 61]. The B20 measurements show good linear correlation (correlation coefficient at 0.7723) with the GC2518 measurments [59].

Used for the measurements of atmospheric $\beta$ ray from 29 May to 1 June 2011, after the Fukushima earthquake on 11 Mar 2011 and the FDNPS accident [62]. Monitoring the radiation situation in the Chernobyl exclusion zone [63]. Surveying some of the most radioactive places on Earth at the end of 2014 [64, 65]. The air dose rates were 1.94 $\mu$Sv/h at Pripyat Hospital, Chernobyl; 0.15 $\mu$Sv/h in Sydney, Australia; 0.30 $\mu$Sv/h in Peace Dome, Hiroshima, Japan; 1.65 $\mu$Sv/h in Jachymov, a mine where uranium was first discovered, in Czech Republic; 1.50 $\mu$Sv/h in the doornub of a laboratory of Marie Curie in Radium Institute, Paris; 0.80 $\mu$Sv/h in the Trinity Bomb Test Site, where the world's first nuclear bomb was tested, in New Mexico. The trinitite found at the Trinity test sites were measured at 2.10 $\mu$Sv/h; In an commercial airplace, the air dose rates of 0.53 $\mu$Sv/h were measured at 18,000 feet altitude, 0.93 $\mu$Sv/h at 23,000 feet, and 2.23 $\mu$/h at 33,000 feet, and 3.00 $\mu$Sv/h at higher altitudes and toward the pole. 5.17 $\mu$Sv/h at the Chernobyl Number 4 reactor; 1.63–10$\mu$Sv/h in Fukushima City (3 years after the accident); 489 $\mu$Sv/h right outside of the Pripyat Hostical, and over 1950 $\mu$Sv/h at the basement of the same Pripyat Hospitcal and over the clothing disposed from the firemen [64].

Measurements of ambient air dose rates from plutonium and other radionuclides sixty years after nuclear tests in the Montelbello Islands, Western Australia [66]. Measurements of ambient air dose rates on the effectiveness of composite materials designed for shielding ionizing radiation from radioative sources [67]. Measurements of ambient air does rates surrounding patients who were injected with radiotive materials for nuclear medicine examinations in the hospitals [19]. Measuring the background radiation levels in the radiology deparment of a hospital were measured between 0.11 $\mu$Sv/h and 0.13 $\mu$Sv/h [68]. Measuring the background radiation levels in from radioactive materials used in the machines in nuclear cardiology field in the hospitcals [69].

Used as one of the 93 instruments at 43 sites over the Nordic countries to quantify the uptake of 131I in thyroids in emergency situations such as nuclear accidents (e.g., Windscale in 1957, Three Mile Island in 1979, Chernobyl in 1986, Fukushima in 2011). This study concludes that various instruments can be used in emergency situation as long as the instruments are properly calibrated. This finding is consistent with the method we used in this work, i.e., the extensive calibrations of all 10 air dose meters during the studying period [20]. Used as one of the two radiation meter used by the Radiation Protection Officer (RPO) in radiation safety in Dublin City University, Ireland [70]. RadEye G (similar to G-10 but with onboard data logger and Windows interface) was used in mobile robot when searching for radiologival source [71]. Used in a drone for measuring $\gamma$ radiation [72]. Used to measure indoor and outdoor $\gamma$ dose rates in major commerical building material distribution outlets [73]. Used in a small and low cost groundrobot for nuclear decommissioning detection [74]. RadEye GN+ is used in an exam paper in a robotics course [75]. A Thermo Fischer RadEye G-10 personal gamma dosimeter with a laptop for communicating with the dosimeter is used in a robotic platform. The G-10 provides radiation measurements to the robotic platform to autonomously navigate through the environment [76].

The radionuclides released from the damaged nuclear reactors in FDNPS are mainly $^{131}I$, $^{132}Te$, $^{134}Cs$, $^{136}Cs$, and $^{137}Cs$ [77, 78]. Radionuclides $^{137}Cs$ from past tests of nuclear weapons were detected at the radionuclide stations from the Comprehensive Nuclear-Test-Ban Treaty (CTBT) Organization (CTBTO) [46, 79]. Radionuclides $^{137}Cs$ have high particle adsorption affinities and were deposited as fallout [66, 78]. Condensable nuclear debris are in the forms of particulates with diameters between 0.01 and 1 $\mu$m [80]. The deposition velocities are 0.0035

cm $s^{-1}$ for particulates with diameter of 1 $\mu$m [81]. Events such as the long-range transport of the Asian dust storms have frequently occurred over the North Pacific [44]. Hence, resuspension of the deposited radioactive particulates are possible for global dispersion of $^{137}Cs$ from past nuclear explosions.

Due to the tests of nuclear weapons in the past, it is necessary to discuss whether it is the effect of the Fukushima or the effect from the continent due to atmospheric nuclear weapons tests over the North Pacific during the period 2011–2015. In order to assess the effects of the resuspension of the radionuclides from the pass tests of nuclear weapons, we have referred to the radionuclide data reported in the annual report published by the CTBTO during the period 2000–2020 [46, 82–100].

Detailed measurements of the radionuclides associated with the fission processes during the period 2000 to 2020 were reported in the CTBTO annual report. The CTBTO radionuclide data were used in verifying model simulations of the global dispersions of the radionuclides from the FDNPS accident [101, 102].

CTBTO reported measurements of the radionuclide data from radionuclide stations around the world. Each station sends one gamma ray spectrum per day. The gamma ray spectrum reveals detailed information on the radionuclides and the quantity [24]. These data were grouped into 5 levels. The first two levels of the radionuclides occur naturally. Level 3 radionuclides are fission products typical for the station detected. Level 4 events contain one CTBT-relevant products detected. Level 5 events contain multiple CTBT-relevant fission products detected.

There are legacy radionuclides produced from previous tests of nuclear weapons [46]. Of all these radionuclides that are capable of long-range transport, $^{137}Cs$ is a prominent radionuclide that has been observed by the CTBTO sites. $^{137}Cs$ is also a prominent radionuclide released from the FDNPS accident [77, 78]. How we tell the $^{137}Cs$ released from the FDNPS accident to those from the tests of the nuclear weapons in the past?

Fig 14 shows locations of tests of nuclear weapons (blue stars) during the period 1945–1998 [103]. Also shown in Fig 14 are 8 radionuclide monitoring stations (with red color coded capital letters) from the CTBTO that have long-period of radionuclide data reported in the CTBTO annual reports. These 8 radionuclide stations are also located downwind of the continental nuclear test sites over central Eurasia and North America. The radionuclide monitoring station Ulaanbaatar (U) is the closest monitoring station to the central Eurasia content nuclear test sites. Takasaki (T) is the closest monitoring station to the FDNPS (at about 150 km distance southwest to the FDNPS). The Reykjavik (R) monitoring station in the North Atlantic is located downwind of North America nuclear test sites. Okinawa (O), Guam (G), and Wake Island (W) are located in the western North Pacific. Midway Island (M) and Hawaii (H) are located in the eastern North Pacific.

Previous studies have shown the impacts of the long-range transport of the Asian dust storm [44], and the downstream transport of the radionuclides from the FDNPS accident [34, 101, 102] over the North Pacific atmosphere. Hence, deposited and soil-absorbed of the long-lived radionuclides such as $^{137}Cs$ [66] are conducive for making longer lasting impacts over the North Pacific. What matter now is the dose from the legacy nuclear tests that can have over the North Pacific atmosphere. Fig 15 shows a time-series plot of the nuclear tests during the period 1945–2019 [104]. The total annual tests of nuclear weapons peaked at 178 tests in 1960. Almost the tests of the nuclear weapons have stopped after 1998.

Atmospheric radioactivity at the sites of nuclear explosions decrease with time. The first UK test of nuclear weapon was conducted on 3 October 1952 at the Monte Bello Islands in western Australia [105]. On 16 May 1956, a 15-kiloton TNT (trinitrotoluene)-equivalent scale of nuclear weapon exploded at the G1 site on Trimouille Island. On 19 June 1956, another

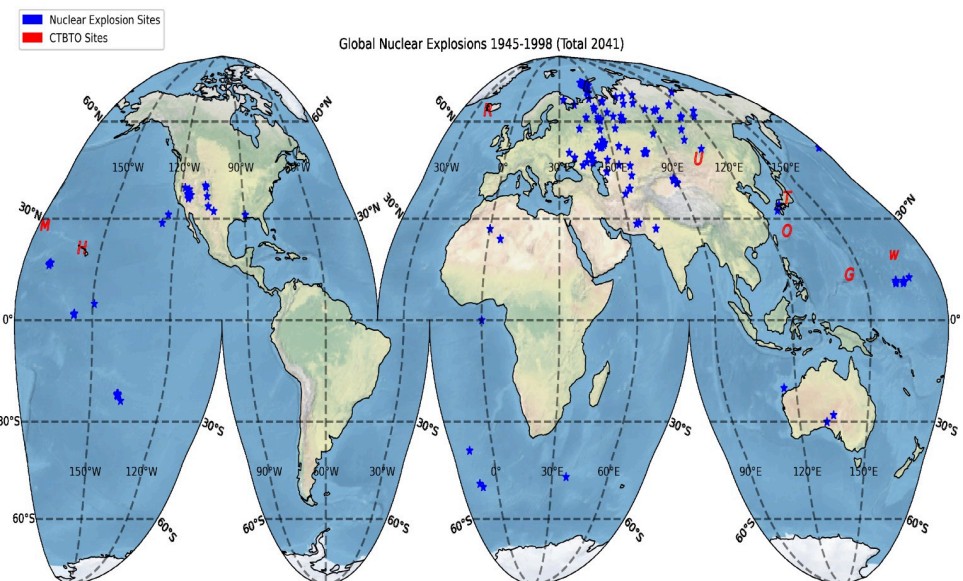

**Fig 14. Global nuclear explosion sites during the period 1945–1998 (blue), and the CTBTO monitoring sites (red) discussed in this work.** Capital red letters indicate names and locations of following sites: U (Ulaanbaatar, 47.9˚N, 106.3˚E); R (Reykjavik, 64.1˚N, 21.9˚W); T (Takasaki, 36.3˚N, 139.1˚E); O (Okinawa, 26.5˚N, 127.9˚E); W (Wake Island, 19.3˚N, 166.6˚E); M (Midway Island, 28.2˚N, 177.4˚W); H (Hawaii, 21.5˚N, 158.0˚W); G (Guam, 13.6˚N, 144.9˚E). Map made with Natural Earth. Free vector and raster map data at naturalearthdata.com.

60-kiloton TNT-equivalent scale of nuclear weapon exploded at the G2 site on Alpha Island. For comparisons, the 20-kiloton TNT equivalent scale of nuclear explosion on 6 August 1945 at Hiroshima, and on 9 August 1945 at Nagasaki [106]. The air dose rates measured at the epicenter of the G1 explosion site is 35 $\mu$Sv/h in 1962, and 0.1 $\mu$Sv/h in 2015 [66]. The air dose rates measured at the epicenter of the G2 explosion sites is 160 $\mu$Sv/h in 1962, and 0.2 $\mu$Sv/h in

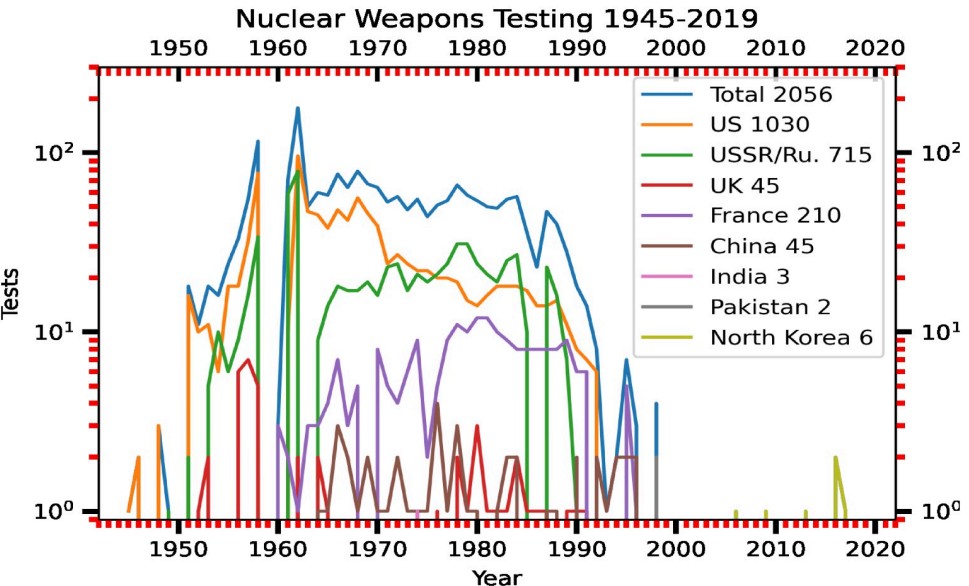

**Fig 15. Annual tally of testings of nuclear weapons with respect to countries during the period 1945–2019.**

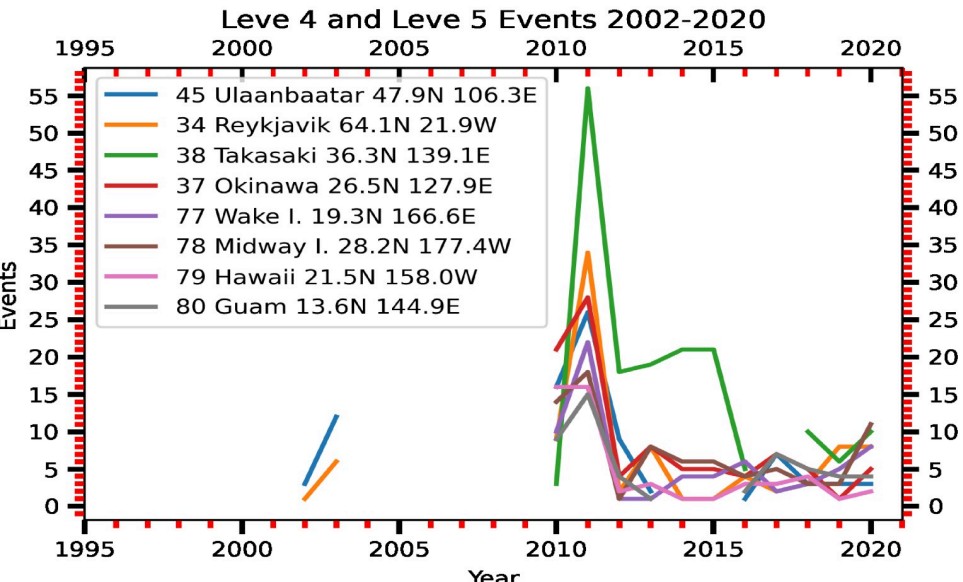

**Fig 16. Annual total of the level 4 and leve 5 events during the period 1945–2019.** The number in front of the station name indicates the radionuclide station index according to the CTBTO.

2015 [66]. Significant drops of the air dose rates in 2015 to 1–3% of the magnitudes measured in 1962 in 53 years. The air dose rates with respect to $^{134}Cs$ and $^{137}Cs$ at the Nagasaki hypocenter is about 0.0042 -0.0081 $\mu$Sv/h as measured in June 2016 [107].

Fig 16 shows time-series plots of the annual sum of the level 4 events and level 5 events that been observed at each of the 8 CTBTO sites, shown in Fig 14, during the period 2002–2020 [46, 82–100, 108]. Notice that there are no data of the sum of level 4 and level 5 events reported in the CTBTO annual report during the period 2004–2009. For the period 2004–2009, the data of the number of occurrences of the CTBT-relevant radionuclides are reported in the annual report for years 2004 [84], 2006 [84], 2007 [84], 2008 [84], and 2009 [84], and is shown in Fig 17. No data of the number of occurrences of CTBT-relevant radionuclides are reported in years 2000 [46] and 2001 [46].

With the data shown in Figs 16 and 17 we can compare and rank (from high to low) the number of occurences of the radionuclids at monitoring stations before and after the FDNPS accident. We first focus on the comparisons of the three radionuclide monitoring stations at Ulaanbaatar (downwind of the central Eurasia content nuclear test sites), Takasaki (located at about 150 km southwest to the FDNPS site), and Reykjavik (downwind of North America nuclear test sites) (Fig 14).

Before the FDNPS accident: Ulaanbaatar > Reykjavik > Takasasi (2002 (U>R), 2003 (U>R), 2004, 2006 (U>T), 2007 (U>R), 2008, 2009, 2010). We call this as a URT pattern. After the FDNPS accident: in 2011: Takasasi > Reykjavik > Ulaanbaatar (TRU pattern); in 2012: Takasaki > Ulannbaatar > Reykjavik (TUR); in 2013: Takasaki > Reykjavik > Ulaanbaatar (TRU pattern); in 2014: Takasaki > Reykjavik (no data from Ulaanbaatar; TR); in 2015: Takasaki > Reykjavik (no data from Ulaanbaatar; TR); in 2016: Takasaki > Reykjavik > Ulaanbaatar (TRU pattern); in 2017: Ulaanbaatar > Reykjavik (no data from Takasaki; UR); in 2018: Takasaki > Ulaanbaatar and Reykjavik (TR and TU); in 2019: Reykjavik > Takasaki > Ulaanbaatar (RTU); in 2020: Takasaki > Reykjavik > Ulaanbaatar (TRU pattern).

Hence, the patterns of the background radionuclides inherited from the legacy tests of nuclear weapons (Figs 14 and 15) are in the orders of URT, UR, and UT. Takasaki site has

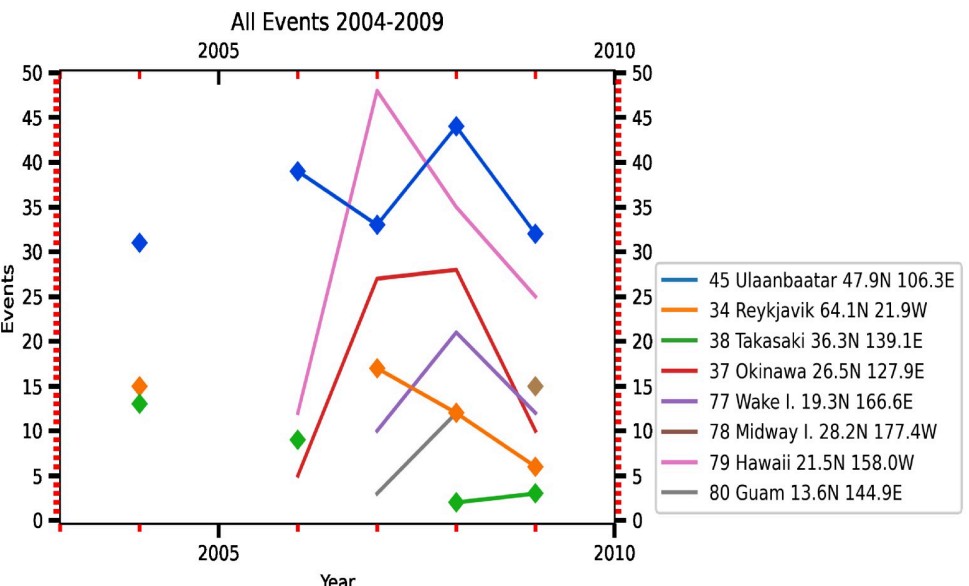

**Fig 17. Annual total events during the period 2004–2009.** The number in front of the station name indicates the radionuclide station index according to the CTBTO.

always had the lowest events of occurrences of radionuclides before the FDNPD accident. In 2011, the pattern has changed to TRU. The TRU patter has occurred in 2011, 2013, 2016, and 2020. Variants of TRU are TUR (2012), RTU (2019), TR (2014, 2015, 2018), UR (2017). Takasaki has never had the lowest events of the occurrences of the radionuclides after the FDNPS accident and during the period 2011–2020.

For the stations Okinawa Island (26.5˚N) and Hawaii (21.5˚N): before the FDNPS accident: Hawaii > Okinawa (2006, 2007, 2008, 2009, 2010); after the FDNPS accident: Hawaii < Okinawa (each year of 2011 to 2020). For the stations Midway Island (28.2˚N) and Hawaii (21.5˚N): before the FDNPS accident: Hawaii > Midway Island (2009, 2010); after the FDNPS accident: Hawaii < Midway Island (each year of 2011 to 2020, except 2012 and 2018). For the stations Wake Island (19.3˚N) and Hawaii (21.5˚N): before the FDNPS accident: Hawaii > Wake Island (2007, 2008, 2009, 2010); after the FDNPS accident: Hawaii > Wake Island (2012, 2013, 2017, 2018), Hawaii < Wake Island (2011, 2014, 2015, 2016, 2019, 2020) For the stations Guam Island (13.6˚N) and Hawaii (21.5˚N): before the FDNPS accident: Hawaii > Guam (2007, 2008, 2010); after the FDNPS accident: Hawaii > Guam (2011, 2013, 2016); Hawaii < Guam (2012, 2017, 2018, 2019, 2020) No data from Guam in 2014 and 2015.

Above comparisons show significant and consistent changes in the number of occurrences of the radionuclides at paired island monitoring sites before and after the FDNPS accident. Before the FDNPS accident, the number of occurrences of the radionuclides at Hawaii are higher than the island stations. After the FDNPS accident, the number of occurrences of the radionuclides at island stations are mostly higher than the Hawaii measurements. These results are consistent with the shifted patterns found in the previous comparisons of Ulaanbaatar, Takasaki, and Reykjavik stations. Hence, the atmosphere over the North Pacific clearly impacted by the radionuclides from the FDNPS accident.

The CTBTO data has also revealed the impact of the FDNPS accident on a global scale as measured by the high resolution of the gamma spectrometers. Fig 18 shows a time-series plot of the total number of the CTBTO stations reporting measurements of the radionuclide data, and the level 5 events during the period 2010–2020. A prominent peak of 41 monitoring

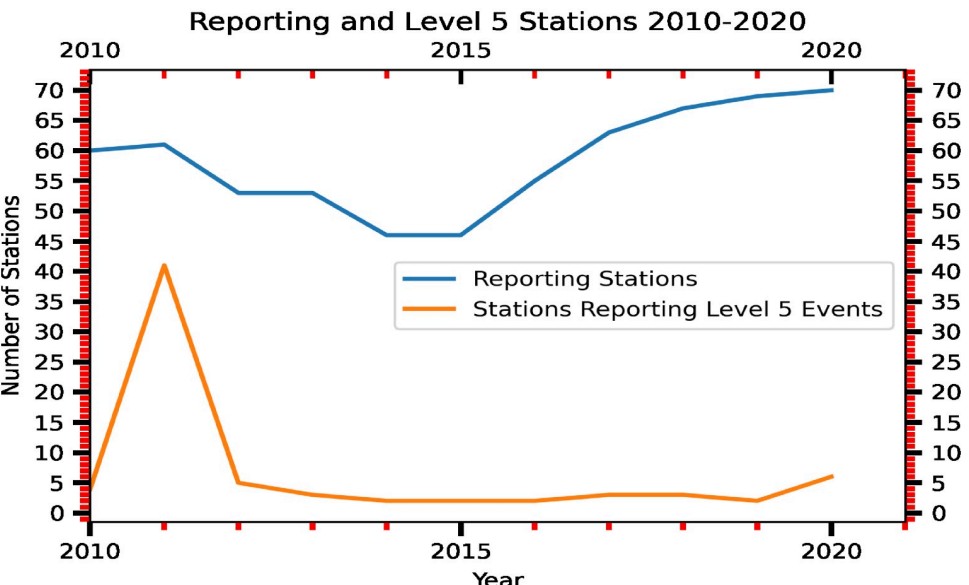

**Fig 18. Annual total stations reporting level 5 events, and total monitoring stations during the period 2010–2020.**

stations reporting level 5 events in 2011 globally. This is about 67% of the total 61 monitoring stations in 2011. Before the FDNPS accident, only 4 monitoring stations reporting the level 5 events in 2010. After the FDNPS accident and during the period 2013 and 2019, about 2 to 4 stations reporting the level 5 events.

Fig 19 shows time-series of the percentage occurrences the CTBT relevant radionuclides observed during the period 2004–2020. Before the FDNPS accident, most of the radionuclides are sodium-24 ($^{24Na}$), $^{137}Cs$, $^{60}Co$, $^{99}MTc$, and $^{131}I$. $^{24}Na$ has a half-life of about 15 hours, and is

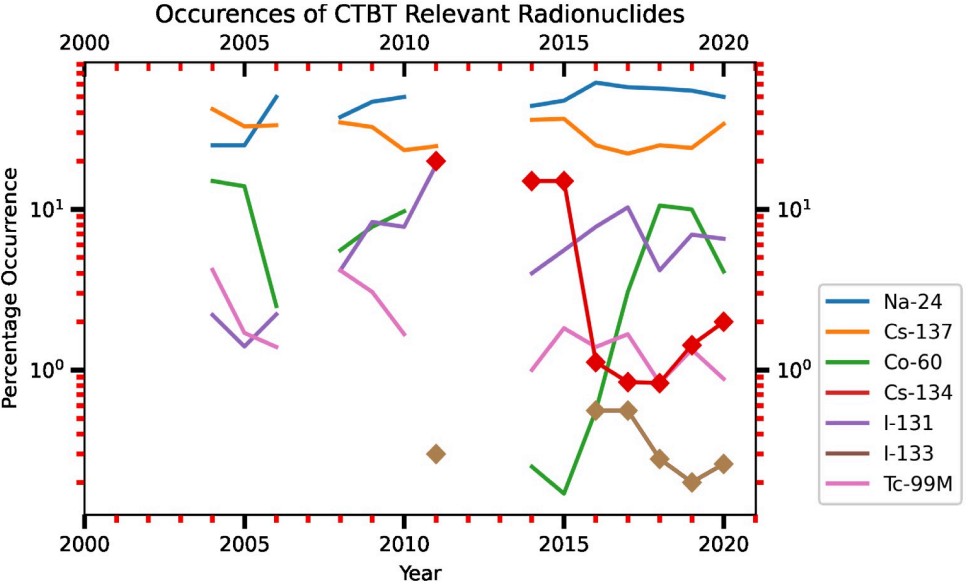

**Fig 19. Annual percentage occurrences of the CTBT-relevant radionuclides during the period 2004–2020.**

mainly used as a medical tracer to trace to cellular uptake of sodium in the body [109]. $^{99}MTc$ is also used as a medical tracer in the whole-body imaging procedures in nuclear machines for diagnosing diseases in brain, thyroid, lungs, liver, gallbladder, kidneys, skeleton, blood, tumors, etc. [110]. $^{60}Co$ has a half-life of 5.27 years and contains high-intensity gamma-ray emitted energy at 1.3 MeV. $^{60}Co$ is widely used as a a tracer in chemical reaction, radiation sources for medical radiotherapy (e.g., to treat cancer), industrial radiography, pest inset sterilization, food irradiation, blood irradiation, etc. [111].

After the FDNPS accident, the percentage occurrences of $^{134}Cs$ are 20% in 2011, about 15% during 2014 and 2015, and between 0.8% and 2% during the period 2016–2020. The percentage occurrences of $^{133}I$ varies between 0.2% and 0.6% during the period 2011–2022 (no data in 2012, 2013, 2014, 2015). Both $^{137}Cs$ (half-life 30.17 years) and $^{134}Cs$ (half-life 2.0648 years) are fission products of uranium and plutonium in nuclear reactors and nuclear weapons [112]. Presence of $134Cs$ radionuclides after the FDNPS accident are consistent with the reported releases of radionuclides from the nuclear reactor of the FDNPS [77, 78]. $^{133}I$ (half-life 20.5 hours) radionuclides are suitable fission indicators [113] or failure in coolant failure in the reactor of nuclear power plants [114]. Notice that 5.1% of $^{132}Te$, 0.7% of $^{95}Nb$, 1.3% of $^{140}La$, 6.4% of $^{136}Cs$, and 22.8% of other radionuclides are reported in CTBTO 2011 annual report [91]. The measurements of radionuclides after the FDNPS accident show the impacts of the FDNPS accident downwind over the North Pacific and spread to a global domain [101, 102].

## Conclusions

In this work, we report the continuous monitoring of air dose rates over the Port of Tokyo, the North Pacific Ocean, and the northwestern Atlantic Ocean after the FDNPS accident. The data were collected from 19 March 2011 to 2 September 2015. To the best of our knowledge, our data are the first continuous measurements of air dose rates taken in the oceanic atmosphere after a significant accident at a nuclear power plant. We collected 248,864 measurements of the oceanic atmosphere at 41,477 locations from 294 cruises. The observed air dose rates were 0.069 (25th percentile), 0.079 (50th percentile), 0.082 (mean), 0.089 (75th percentile), 0.099 (90th percentile), and 1.03 (maximum) $\mu Sv/h$.

The air dose rates measured over the land surface at the Port of Tokyo and the measurements made over the ocean surface on the calling ships berthed at the Port of Tokyo were impacted by the FDNPS emissions. The time-series measurements of the air dose rates of the North Pacific atmosphere made by oceanic sailing ships were impacted by radioactive plumes transported downwind to over the North Pacific Ocean and over the northwestern Atlantic Ocean. Hence, our data confirm the global impact of the radionuclides from the FDNPS in the North Pacific atmosphere and the northwestern Atlantic atmosphere as predicted by, for example, Povinec et al. [33].

All the calibrated data files reported in this work are available at http://doi.org/10.5281/zenodo.3563214. The data we submitted are reachable with one click (without the need for a login and password) and a second click to download the data, consistent with the two-click access principle for data published in Carlson and Oda [115].

## Supporting information

**S1 Appendix.**
(PDF)

## Acknowledgments

We are very grateful to the Evergreen Marine Corporation (EMC) for participating in the PGGM/IAGOS project and for supporting the B-20 and the G-10 sensors used in this work. We dedicate this work to the EMC founder, the late Dr Yung-Fa Chang, for his visionary support of the PGGM project. We thank Tang-Huang Lin of CSRSR for granting us a permission to use Spot 6 satellite image in this work. We dedicate this work to those who have suffered from the FDNPS accident. We thank the following container ships: Ever Uranus, Ever United, Ever Unicorn, Ever Useful, Union, Ever Uberty, Ever Unific, Ever Utile, Ever Urban, Ever Ultra, Ever Utile, Ever Union, Ever Ulysses, Ever Ursula, Ever Unison, Ever Salute, Ever Strong, Ever Smile, Ever Safety, Ever Summit, Ever Sigma, Ever Shine, Ever Dainty, Ever Develop, Ever Decent, Ever Diamond, Ever Delight, Ever Dynamic, Ever Diadem, Ever Ethic, Ever Excel, Ever Elite, Ever Eagle, Ever Envoy, Ever Racer, Ever Radiant, and Ever Respect.

## Author Contributions

**Conceptualization:** Kuo-Ying Wang, Neil Harris.

**Data curation:** Kuo-Ying Wang, Mizuo Kajino, Yasuhito Igarashi.

**Formal analysis:** Kuo-Ying Wang.

**Funding acquisition:** Kuo-Ying Wang.

**Investigation:** Kuo-Ying Wang, Philippe Nedelec, Hannah Clark, Mizuo Kajino, Yasuhito Igarashi.

**Methodology:** Kuo-Ying Wang, Neil Harris, Mizuo Kajino, Yasuhito Igarashi.

**Project administration:** Kuo-Ying Wang.

**Resources:** Kuo-Ying Wang, Philippe Nedelec, Hannah Clark, Neil Harris, Mizuo Kajino, Yasuhito Igarashi.

**Software:** Kuo-Ying Wang.

**Supervision:** Kuo-Ying Wang, Hannah Clark, Yasuhito Igarashi.

**Validation:** Kuo-Ying Wang.

**Visualization:** Kuo-Ying Wang.

**Writing – original draft:** Kuo-Ying Wang.

**Writing – review & editing:** Kuo-Ying Wang, Philippe Nedelec, Neil Harris, Mizuo Kajino, Yasuhito Igarashi.

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
