## [Decision Letter · Decision Letter 0]

29 Mar 2022

PONE-D-21-14834

Impacts on air dose rates after the Fukushima accident over the North Pacific from 19 March 2011 to 2 September 2015

PLOS ONE

Dear Dr. Wang,

Thank you for submitting your manuscript to PLOS ONE. After careful consideration, we feel that it has merit but does not fully meet PLOS ONE’s publication criteria as it currently stands. Therefore, we invite you to submit a revised version of the manuscript that addresses the points raised during the review process.

We look forward to receiving your revised manuscript.

Kind regards,

James R. Lyons, PhD

Academic Editor

PLOS ONE

Journal Requirements:

4. Thank you for stating the following financial disclosure: "KYW is funded by the Ministry of Science and Technology (www.most.gov.tw; 107-2111-M-008-027-). PN and HC are supported by the the IAGOS project (www.iagos.org)."

6. We note that Figure 1 in your submission contain satellite images which may be copyrighted. All PLOS content is published under the Creative Commons Attribution License (CC BY 4.0), which means that the manuscript, images, and Supporting Information files will be freely available online, and any third party is permitted to access, download, copy, distribute, and use these materials in any way, even commercially, with proper attribution. For these reasons, we cannot publish previously copyrighted maps or satellite images created using proprietary data, such as Google software (Google Maps, Street View, and Earth). For more information, see our copyright guidelines: http://journals.plos.org/plosone/s/licenses-and-copyright.

Additional Editor Comments (if provided):

Please add a thorough discussion of the suitability of the GM counter for the measurements reported in your manuscript. Include any technical data that support the use of the GM counter for this work, and also cite and discuss other examples in which comparable measurements have been made with this instrument.

Reviewers' comments:

Reviewer's Responses to Questions

**Comments to the Author**

1. Is the manuscript technically sound, and do the data support the conclusions?

Reviewer #1: Yes

Reviewer #2: No

2. Has the statistical analysis been performed appropriately and rigorously? 

Reviewer #1: I Don't Know

Reviewer #2: Yes

3. Have the authors made all data underlying the findings in their manuscript fully available?

Reviewer #1: Yes

Reviewer #2: Yes

4. Is the manuscript presented in an intelligible fashion and written in standard English?

Reviewer #1: Yes

Reviewer #2: Yes

5. Review Comments to the Author

Reviewer #1: General comments

The authors present the first continuous measurements of air dose rates in the ocean atmosphere after a major accident at a nuclear power plant. The data is considered to be very valuable. The amount of data is shown by many figures, but it was difficult to read many supplementary figures, especially in the discussion. I think it would have been easier to understand the discussions if they were presented together with the results. In terms of the suspension, I think it is necessary to discuss whether it is the effect of the Fukushima accident or the effect from the continent due to atmospheric nuclear weapons tests

I believe that this paper is suitable for the publication in PLOS ONE after minor revisions.

Specific comments

L36 The PGGM project needs to be cited here.

Figure 1 and 2; The earth quake epicenter is not mentioned in the text, so I don't think it needs to be in this diagram.

Ｌ150 The authors should describe the model domain in an easy-to-understand manner in terms of latitude and longitude.

L171 The month of March 2011 is a period of large-scale atmospheric depression. Fig. 3(a) shows one month's data at a time, but a more detailed analysis is needed.

L213 The authors should include Fig. S2 in the text, as it is an important result. The results of the statistical analysis should also be described in more detail.

L245 Aoyama et al., 2016 is not in References; please cite 31 and 40 more correctly. Also, there doesn't seem to be any elevated Cs-137 measurements for this period in the cited sources.

L251 It is difficult for me to find out that the months of increase in the air dose rates mostly occur from November to April and from June to August in Fig 4 e. The authors should these periods more clearly.

L399 I think it is very unfortunate that the comparison is limited to Tokyo Bay only, despite the availability of data for the Pacific Ocean. For example, one of the authors has developed the North Pacific scale model presented in (Aoyama et al., 2016). I think a comparison with the North Pacific Scale Model would be very valuable.

L503 In terms of resuspension, I think it is necessary to discuss whether it is the effect of the Fukushima accident or the effect from the continent due to atmospheric nuclear weapons tests.

Reviewer #2: The paper presents results of unique network of dose rate measurements done all over the world on board on merchant ships. However the interpretation, especially that regarding connection of North Atlantic results with post-accident trace releases from Fukushima seems to be for me not enough supported. The GM counter, as only dose rate meter is not sufficient tool for such long distance study.

6. PLOS authors have the option to publish the peer review history of their article (what does this mean?). If published, this will include your full peer review and any attached files.

Reviewer #1: No

Reviewer #2: No

---

## [Author Response · Author response to Decision Letter 0]

28 May 2022

Response to reviewers’ comments. Manuscript Number PONE-D-21-14834

Impacts on air dose rates after the Fukushima accident over the North Pacific from 19 March 2011 to 2 September 2015

By Wang et al.

We are very grateful indeed to the reviewers, the editor, and the editorial office for very encouraging and insightful comments that have significantly increased the clarity of the manuscript. We have revised the manuscript following the comments. A detailed point to point response to each comment is given below.

* A rebuttal letter that responds to each point raised by the academic editor and

reviewer(s). You should upload this letter as a separate file labeled 'Response to

Reviewers'.

Reply. We have uploaded the Response to Reviewers.

* A marked-up copy of your manuscript that highlights changes made to the original

version. You should upload this as a separate file labeled 'Revised Manuscript with Track Changes'.

Reply. We have uploaded a separated file, labeled “Revised_Manuscript_with_Track_Changes.pdf”.

Reply. We have uploaded an unmarked version of the revised manuscript.

Journal Requirements:

1. Please ensure that your manuscript meets PLOS ONE's style requirements, including 

those for file naming. The PLOS ONE style templates can be found at 

https://journals.plos.org/plosone/s/file?id=wjVg/PLOSOne_formatting_sample_main_body.pdf and https://journals.plos.org/plosone/s/file? id=ba62/PLOSOne_formatting_sample_title_authors_affiliations.pdf

Reply. We have reformatted the manuscript following the PLOS LaTex template.

2. Please update your submission to use the PLOS LaTeX template. The template and more information on our requirements for LaTeX submissions can be found at 

http://journals.plos.org/plosone/s/latex.

Reply. We have submitted the revised manuscript using the PLOS LaTex template.

3. We note that the grant information you provided in the ‘Funding 

Information’ and ‘Financial Disclosure’ sections do not match.

Reply. We have provided grant number in the Acknowledgements.

4. Thank you for stating the following financial disclosure: "KYW is funded by the 

Ministry of Science and Technology (www.most.gov.tw; 107-2111-M-008-027-). PN and HC are supported by the the IAGOS project (www.iagos.org)."

Reply. Thank you very much for your confirmation.

Reply. We have included this Role of Funder statement in the cover letter.

5. We note that you have stated that you will provide repository information for your 

data at acceptance. Should your manuscript be accepted for publication, we will hold it until you provide the relevant accession numbers or DOIs necessary to access your data. If you wish to make changes to your Data Availability statement, please describe these changes in your cover letter and we will update your Data Availability statement to reflect the information you provide.

Reply. We have described in the manuscript that all the data of air dose rates measurements reported in this work are openly available at http://doi.org/10.5281/zenodo.3563214.

6. We note that Figure 1 in your submission contain satellite images which may be 

copyrighted. All PLOS content is published under the Creative Commons Attribution 

License (CC BY 4.0), which means that the manuscript, images, and Supporting Information files will be freely available online, and any third party is permitted to access, download, copy, distribute, and use these materials in any way, even commercially, with proper attribution. For these reasons, we cannot publish previously copyrighted maps or satellite images created using proprietary data, such as Google software (Google Maps, Street View, and Earth). For more information, see our copyright guidelines: http://journals.plos.org/plosone/s/licenses-and-copyright.

We require you to either (1) present written permission from the copyright holder to 

publish these figures specifically under the CC BY 4.0 license, or (2) remove the 

figures from your submission:

Reply. We have replaced the satellite image used in Fig. 1 with a satellite image from CSRSR. We have obtained a written permission from the Director of CSRSR for using this satellite in the manuscript.

We recommend that you contact the original copyright holder with the Content Permission Form (http://journals.plos.org/plosone/s/file?id=7c09/content-permission-form.pdf) and 

the following text: “I request permission for the open-access journal PLOS ONE to publish XXX under the Creative Commons Attribution License (CCAL) CC BY 4.0 (http://creativecommons.org/licenses/by/4.0/). Please be aware that this license allows unrestricted use and distribution, even commercially, by third parties. Please reply and provide explicit written permission to publish XXX under a CC BY license and complete the attached form.”

Please upload the completed Content Permission Form or other proof of granted 

permissions as an "Other" file with your submission.

Reply. We have uploaded the completed Content Permission Form signed by CSRSR.

In the figure caption of the copyrighted figure, please include the following text: 

“Reprinted from [ref] under a CC BY license, with permission from [name of 

publisher], original copyright [original copyright year].”

Reply. We have included the copyright text in the Figure 1 caption.

7. Please include captions for your Supporting Information files at the end of your 

manuscript, and update any in-text citations to match accordingly. Please see our 

Supporting Information guidelines for more information: 

http://journals.plos.org/plosone/s/supporting-information.

Reply. We have included captions of Supporting Information files at the end of the manuscript.

Additional Editor Comments (if provided):

Please add a thorough discussion of the suitability of the GM counter for the 

measurements reported in your manuscript. 

Reply. We have included a thorough discussion of the suitability of the GM counter for the measurements reported in this work in the section Discussion. 

Include any technical data that support the use of the GM counter for this work, and also cite and discuss other examples in which comparable measurements have been made with this instrument.

Reply. The GM counters RadEye B20 and G10 were used in this work. Both GM counters have been extensively used in published works related to the measurements of air dose rates in places associated with tests of nuclear weapons, accidents from nuclear power plants, medical imaging processes with patients injected with nuclear materials, robotic exploration of nuclear environment, radioactivity safety training, etc. 

We have included technical data from above examples that support the use of the GM counter for this work. We have cited and discussed above examples in which comparable measurements have made with the GM counter used in this work.

Reviewer #1: General comments

The authors present the first continuous measurements of air dose rates in the ocean 

atmosphere after a major accident at a nuclear power plant. The data is considered to be very valuable. The amount of data is shown by many figures, but it was difficult to read many supplementary figures, especially in the discussion. I think it would have been easier to understand the discussions if they were presented together with the results. In terms of the suspension, I think it is necessary to discuss whether it is the effect of the Fukushima accident or the effect from the continent due to atmospheric nuclear weapons tests I believe that this paper is suitable for the publication in PLOS ONE after minor revisions.

Reply. We are very grateful to the reviewer for very insightful comments that have significantly increased the clarity of the manuscript. We have followed reviewer’s comments to revised the manuscript. In the following, we provide point-to-point responses to reviewer’s comments.

Specific comments

L36 The PGGM project needs to be cited here.

Reply. We have included a citation to the PGGM project.

Figure 1 and 2; The earth quake epicenter is not mentioned in the text, so I don’t think it needs to be in this diagram.

Reply. We have included following description in the text:

“(see Fig. 1 for the epicenter of the 11-March-2011 earthquake and the FDNPS site)”

Ｌ150 The authors should describe the model domain in an easy-to-understand manner in terms of latitude and longitude.

Reply. We have included following description of the model domain:

“The model has a domain area from $125^{\\circ}E$ to $155^{\\circ}E$ in longitudes, and from $28^{\\circ}N$ to $48^{\\circ}N$ in latitudes.”

L171 The month of March 2011 is a period of large-scale atmospheric depression. Fig. 

3(a) shows one month’s data at a time, but a more detailed analysis is needed.

Reply. We have included following analysis in the description:

“We note that the month of March 2011 is a period of large-scale atmospheric depression. Two large-scale low pressure systems are located on both side of the Bering Sea: The Aleutian Low on the left, and on the right is a surface low pressure system over the off-shore of area between the Bering Sea and the western Pacific coast of Canada.”

L213 The authors should include Fig. S2 in the text, as it is an important result. The 

results of the statistical analysis should also be described in more detail.

Reply. We have followed reviewer’s comment to include Fig. S2 in the main text, and include a description and a reference of the statistical method used in this work.

L245 Aoyama et al., 2016 is not in References; please cite 31 ad 40 more correctly. 

Also, there doesn’t seem to be any elevated Cs-137 measurements for this period in the cited sources.

Reply. We have corrected citation to Aoyama et al., 2016 (original cite 31). We have checked and corrected citations to 31 and 40. We have revised the description “The dose rates increased again from December 2011 to January 2012, during May-August and in October 2012; in January, March, and June-September 2013,…” (old, lines 241-242)

The Cs-137 measurements shown in Fig. 9 (and also in Fig. 2 of Aoyama et al. 2016 part 1 paper). There exists local maximum of Cs-137 of about 1.2E+04 Bq/m3 relative to nearby measurements of 0.8-0.9 Bq/m3 during December 2011 – January 2012. During May-August 2012, a local maximum of 0.9E+04 Bq/m3. In October 2012, a local maximum of 0.8E+04 Bq/m3. In January 2013, a local maximum of 1.6E+03 Bq/m3. In March 2013, a local maximum of 2.1E+03 Bq/m3. During June-September 2013, a local maximum of 5.5E+03 Bq/m3. During July-October 2014, a local maximum of 3.1E+03 Bq/m3.

We have revised the following description:

“The dose rates increased again from December 2011 to January 2012, during

May-August and in October 2012; in January, March, and June-September 2013; and in July-October 2014. Importantly, these are also periods showing existence of a local maximum of elevated $^{137}Cs$ measurements in the surface water of the FDNPS (see Fig.~\\ref{fig10} and in Fig. 2 of Aoyama et al., 2016$^{31}$).”

L251 It is difficult for me to find out that the months of increase in the air dose 

rates mostly occur from November to April and from June to August in Fig 4 e. The 

authors should these periods more clearly.

Reply. We have included a schematic figure to highlight the months of increase in the air dose rates from time-series plot shown in Fig 4e.

L399 I think it is very unfortunate that the comparison is limited to Tokyo Bay only, 

despite the availability of data for the Pacific Ocean. For example, one of the authors 

has developed the North Pacific scale model presented in (Aoyama et al., 2016). I think a comparison with the North Pacific Scale Model would be very valuable.

Reply. We are very grateful to the reviewer for suggesting the use of the dose rate data over the North Pacific for verifying model results. The initial comparisons over the Tokyo Bay is encouraging. Previous modeling studies compared model results with the CTBTO data over the North Pacific. We are working on the follow-up work to compare model simulations with the air dose rates over the North Pacific. This would be very valuable, as commented by the reviewer. 

Our measurements of the air dose rates over the land and over the water shows quite different characteristics. The mechanism of resuspension, as shown in Kajino et al. (2016), will be a key factor for assessing model capabilities in simulating the dispersions of the radionuclides after the FDNPS accident.

L503 In terms of resuspension, I think it is necessary to discuss whether it is the effect of the Fukushima accident or the effect from the continent due to atmospheric nuclear weapons tests.

Reply. We have added discussion of the effect from the continent due to the atmospheric nuclear weapon tests. We have based our discussion on the radionuclide data collected and analyzed by the gamma spectrometers in the global radionuclide stations from the Comprehensive Nuclear-Test-Ban Treaty (CTBT) Organization (CTBTO) during the period 2000-2020. We have included the discussion using CTBTO data in section discussion.

Reviewer #2: The paper presents results of unique network of dose rate measurements done all over the world on board on merchant ships. However the interpretation, especially that regarding connection of North Atlantic results with post-accident trace releases from Fukushima seems to be for me not enough supported. The GM counter, as only dose rate meter is not sufficient tool for such long distance study.

Reply. We are very grateful to the reviewer for the comment. 

The dose rate measurements over the northwestern Atlantic, as shown in Fig. 8(d) (revised manuscript) were faint, but were still positively correlated with the FDNPS emissions fluxes and the Tsukuba deposition fluxes. In the revised manuscript, we have used data from CTBTO for impacts of the FDNPS. The CTBTO measurements at Reykjavik provides good data from gamma spectrometer measurements of the CTBT-relevant radionuclides. We have included this description in the text.

Reply. We have not found attachment file when logged in and checked for the action link “View Attachments”.

While revising your submission, please upload your figure files to the Preflight 

Analysis and Conversion Engine (PACE) digital diagnostic tool, 

https://pacev2.apexcovantage.com/. PACE helps ensure that figures meet PLOS 

requirements. To use PACE, you must first register as a user. Registration is free. 

Then, login and navigate to the UPLOAD tab, where you will find detailed instructions on how to use the tool. If you encounter any issues or have any questions when using PACE, please email PLOS at figures@plos.org. Please note that Supporting Information files do not need this step.

Reply. We have uploaded all figures to PACE at https://pacev2.apexcovantage.com as suggested. The figure tiff files were checked and downloaded from PACE.

Reply. Thank you very much indeed.

---

## [Decision Letter · Decision Letter 1]

29 Jun 2022

PONE-D-21-14834R1

Impacts on air dose rates after the Fukushima accident over the North Pacific from 19 March 2011 to 2 September 2015

PLOS ONE

Dear Dr. Wang,

Thank you for submitting your manuscript to PLOS ONE. After careful consideration, we feel that it has merit but does not fully meet PLOS ONE’s publication criteria as it currently stands. Therefore, we invite you to submit a revised version of the manuscript that addresses the points raised during the review process.

Please address the minor issues raised by Reviewer 2 to improve manuscript clarity and to correct a couple of typographical errors.

We look forward to receiving your revised manuscript.

Kind regards,

James R. Lyons, PhD

Academic Editor

PLOS ONE

Journal Requirements:

Reviewers' comments:

Reviewer's Responses to Questions

**Comments to the Author**

1. If the authors have adequately addressed your comments raised in a previous round of review and you feel that this manuscript is now acceptable for publication, you may indicate that here to bypass the “Comments to the Author” section, enter your conflict of interest statement in the “Confidential to Editor” section, and submit your "Accept" recommendation.

Reviewer #1: All comments have been addressed

Reviewer #2: All comments have been addressed

2. Is the manuscript technically sound, and do the data support the conclusions?

Reviewer #1: Yes

Reviewer #2: Yes

3. Has the statistical analysis been performed appropriately and rigorously? 

Reviewer #1: Yes

Reviewer #2: Yes

4. Have the authors made all data underlying the findings in their manuscript fully available?

Reviewer #1: Yes

Reviewer #2: Yes

5. Is the manuscript presented in an intelligible fashion and written in standard English?

Reviewer #1: Yes

Reviewer #2: Yes

6. Review Comments to the Author

Reviewer #1: I believe the authors have made appropriate revisions in accordance with the peer review opinion.

Therefore, I believe this paper is suitable for publication in the PLOS ONE.

I look forward to the next developments using the North Pacific scale atmospheric model.

Reviewer #2: Thank you for improving the text by introducing the changes suggested by reviewers. It a bit pity, that during improvements the numeration of rows disappeared from left side, and on right side it is cut out. This makes referring to certain places a bit difficult.

Problematic sites remains:

In a section “The Monitoring Devices and Calibration” there are questions tags left in the brackets with references. This suggests that some reference is missing. In the same section the measurement range is given only using maximum values (< 2mSv/h), whereas the minimum detectable value of 10 microSv/h is given a bit later. So I think the initially maximum value should be named for instance “the maximum detectable dose rate” or “the upper limit of measurement range” not the “range” which apparently was from 10 micro to 2 mili Sv/h.

In a section “Emission and Deposition Fluxes” please note that “chi squared test” is written with small character (“Chi” is not any name)

7. PLOS authors have the option to publish the peer review history of their article (what does this mean?). If published, this will include your full peer review and any attached files.

Reviewer #1: No

Reviewer #2: No

---

## [Author Response · Author response to Decision Letter 1]

17 Jul 2022

Response to reviewers’ comments. Manuscript Number PONE-D-21-14834R1

Impacts on air dose rates after the Fukushima accident over the North Pacific from 19 March 2011 to 2 September 2015

By Wang et al.

We are very grateful indeed to the reviewers, the editor, and the editorial office for very encouraging and insightful comments that have significantly increased the clarity of the manuscript. We have revised the manuscript following the comments. A detailed point to point response to each comment is given below.

Reviewer #2: Thank you for improving the text by introducing the changes suggested by reviewers. It a bit pity, that during improvements the numeration of rows disappeared from left side, and on right side it is cut out. This makes referring to certain places a bit difficult.

Reply. We have revised the line numbers to the left of the manuscript so that the row numbers can be viewed clearly.

Problematic sites remains:

In a section “The Monitoring Devices and Calibration” there are questions tags left in the brackets with references. This suggests that some reference is missing.

Reply. The questions tags were produced due to a missing numbers used in the cited reference which latex cannot find it. We have revised the cited reference numbers in the revised manuscript. (page 3, lines 75-76)

In the same section the measurement range is given only using maximum values (< 2mSv/h), whereas the minimum detectable value of 10 microSv/h is given a bit later. So I think the initially maximum value should be named for instance “the maximum detectable dose rate” or “the upper limit of measurement range”; not the “range” which apparently was from 10 micro to 2 mili Sv/h. 

Reply. We have followed reviewer’s comment to revise the description to “the upper limit of measurement range”. (page 3, lines 74-76)

In a section “Emission and Deposition Fluxes” please note that “chi squared test” is written with small character (“Chi” is not any name)

Reply. We have revised the Chi to chi, following reviewer’s comment. (page 6, line 191)

---

## [Editor Report · Decision Letter 2]

1 Aug 2022

Impacts on air dose rates after the Fukushima accident over the North Pacific from 19 March 2011 to 2 September 2015

PONE-D-21-14834R2

Dear Dr. Wang,

We’re pleased to inform you that your manuscript has been judged scientifically suitable for publication and will be formally accepted for publication once it meets all outstanding technical requirements.

Kind regards,

James R. Lyons, PhD

Academic Editor

PLOS ONE
---

## [Editor Report · Acceptance letter]

3 Aug 2022

PONE-D-21-14834R2 

Impacts on air dose rates after the Fukushima accident over the North Pacific from 19 March 2011 to 2 September 2015 

Dear Dr. Wang:

I'm pleased to inform you that your manuscript has been deemed suitable for publication in PLOS ONE. Congratulations! Your manuscript is now with our production department. 

Kind regards, 

on behalf of

Dr. James R. Lyons 

Academic Editor

PLOS ONE